# Conformal Regression under Distribution Shift: A Reinforcement Learning Method for Adaptive Uncertainty Quantification

## Abstract

Conformal prediction (CP) offers distribution-free uncertainty quantification with formal coverage guarantees, and has been widely applied to regression tasks, including time-series forecasting. However, in time-series settings, the exchangeability assumption underlying CP is often violated due to temporal dependencies. To address this, recent adaptive CP methods mitigate distributional shifts by dynamically calibrating intervals based on recent residuals and adaptive weighting strategies. However, these methods remain limited by their sensitivity to outliers, inability to detect systematic prediction bias, and the decoupling of calibration from model learning. In this work, we introduce CORE that establishes a mutual feedback loop between reinforcement learning (RL) and conformal prediction for adaptive uncertainty quantification. The method leverages RL's exploration capability to better cover uncertain or outlier regions, adapts calibration through exploration feedback, and designs uncertainty-guided rewards, enabling dynamically improved prediction and interval quality through policy interaction and feedback. We conduct extensive experiments to validate its effectiveness across 8 time-series standard datasets. The results demonstrate that our approach achieves superior accuracy and calibration, consistently outperforming 6 state-of-the-art baselines, with an average improvement of 1.36% in coverage rate and 5.03% in interval length.

## 1 Introduction

Conformal prediction (CP) is a model-agnostic framework for distribution-free uncertainty quantification that provides rigorous finite-sample coverage guarantees under the assumption of data exchangeability (De Finetti, 1937; Shafer & Vovk, 2008). In regression task, let $(X, Y) \in \mathbb{R}^d \times \mathbb{R}$ denote the covariate-response pair. Given exchangeable samples $\{(X_i, Y_i)\}_{i=1}^n$ and a target miscoverage level $\alpha \in (0, 1)$, conformal regression constructs a prediction interval $\hat{C}(X_{n+1}) \in \mathbb{R}$ that attains marginal coverage at least $1 - \alpha$: $\Pr(Y_{n+1} \in \hat{C}(X_{n+1})) \geq 1 - \alpha$ (Vovk et al., 2005; Lei et al., 2018). However, in sequential time-series forecasting task, temporal dependencies break the exchangeability assumption, since the joint distribution of the process is not invariant under reordering (Gibbs & Candes, 2021). It makes the reliability of classical conformal regression methods be suboptimal (Lei et al., 2018; Romano et al., 2019; Xu & Xie, 2021).

Recent advances have introduced adaptive or online CP methods (Gibbs & Candes, 2021; Zaffran et al., 2022), which rely on approximate independence assumptions and update intervals through sliding or weighted calibration, often with dynamic adjustments of conformity scores, thereby enabling responsive uncertainty quantification under distribution shifts (Xu & Xie, 2021; Gibbs & Candes, 2021). However, despite the flexibility of adaptive CP methods, they face some fundamental limitations in dynamic environments, as depicted in Figure 1(a). Firstly, they are sensitive to outliers and might fail to capture prediction bias under evolving patterns, leading to inefficient or distorted coverage (Cheung et al., 2024). Secondly, the post-hoc calibration adjustment (Huang et al., 2024) operates independently from model training, limiting feedback that could correct bias or improve uncertainty estimation, and thus reducing robustness under shifts. These challenges motivate a new approach that robustly provides accurate predictions, and integrates calibration with model training to jointly improve coverage and uncertainty estimation.

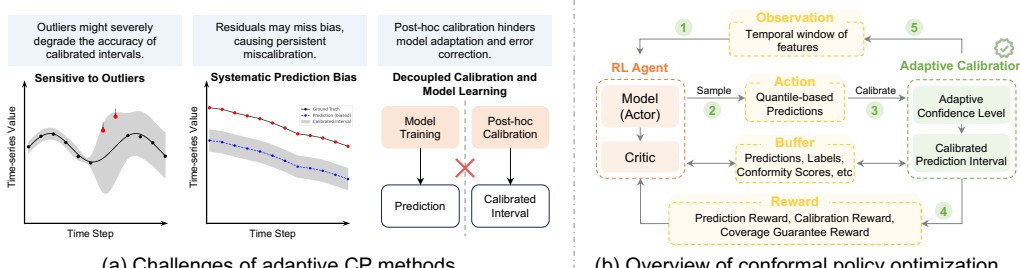

(a) Challenges of adaptive CP methods  (b) Overview of conformal policy optimization

Figure 1: Challenges of adaptive CP methods and our proposed `CORE` architecture.

To this end, we propose `CORE`, standing for **CO**nformal regression with **RE**inforcement learning (RL), which addresses conformal regression under distribution shifts by integrating RL and CP into a dynamic feedback loop for adaptive uncertainty quantification. Instead of treating calibration as a post-hoc correction, our proposed method, as shown in Figure 1 (b), establishes a dynamic feedback loop between the agent, environment, and calibration mechanism, adapting intervals to historical predictions. Concretely, `CORE` formulates conformal time-series prediction as a sequential decision-making problem, where the agent adopts a quantile-guided actor–critic framework to generate prediction intervals. An adaptation-aware calibration module then dynamically updates conformity scores and confidence levels using a sliding calibration buffer, ensuring validity under temporal drift. Finally, uncertainty-guided rewards couple prediction accuracy, coverage validity, and interval efficiency into the RL objective, enabling the policy to improve both prediction and calibration through interaction.

We conduct empirical experiments along 3 complementary tasks, as summarized in Table 1. We evaluate `CORE` on 8 standard time-series datasets for both uncertainty quantification and point prediction, aiming to assess coverage validity, interval efficiency and prediction accuracy. We also extend `CORE` to an anomaly detection variant `CORE-AD` to assess detection performance. Beyond empirical results, we provide theoretical analysis establishing a coverage guarantee for `CORE`. The coverage guarantee is $\sup_{t \leq T} \left| \Pr\{Y_t \in \hat{C}_{1-\alpha}^{(w,t)}\} - (1-\alpha) \right| = \mathcal{O}\left((\log w_c/w_c)^{1/3} + T^{-p/3} + \gamma^{1/2}\right)$, where $T$ is the training set, $w_c$ is the calibration window size, and $p \in [1, 2], \gamma \in [0, 1]$ are constants. This theoretical result ensures `CORE` maintains validity at the confidence level $1 - \alpha$ even under weak dependence and distributional drift.

The main contributions of this work are summarized as follows:

- We propose a mutual feedback loop between RL and CP, enabling the agent to learn uncertainty-aware policies while simultaneously improving calibration through interaction.

- Our empirical results show that `CORE` achieves strong performance across all tasks, with an average improvement of $1.36\%$ in coverage rate and $5.03\%$ in interval length.

- Our theoretical analysis establishes both validity and efficiency of interval lengths in dynamic environments.

## 2 PRELIMINARY

### 2.1 CONFORMAL REGRESSION

Let $(X, Y) \in \mathbb{R}^d \times \mathbb{R}$ with $Y = f(X) + \epsilon$, where $\epsilon$ is independent of $X$. Here $f$ denotes a generic predictive model, and conformal prediction is *model-agnostic*, independent of the specific form of $f$. Given samples $(\mathbf{x}_i, y_i)_{i=1}^n$, a conformity score is introduced to measure how well a candidate pair $(x, y)$ aligns with the predictive model. A common choice in regression is the absolute residual, i.e., $S(x, y) = |\hat{f}(x) - y|$. Regardless of the specific form of the conformity score, the key issue is to determine how small the score must be for $\tilde{f}(x)$ to be accepted as a reasonable prediction (Gibbs & Candes, 2021). For a specified index set $\mathcal{I}_{\text{cal}}$, a held-out calibration set $\mathcal{D}_{\text{cal}} = \{(\mathbf{x}_i, y_i)\}_{i \in \mathcal{I}_{\text{cal}}}$ disjoint from the data is used to fit the regression model (Romano et al., 2019), establishes empirical quantile thresholds on conformity scores for acceptance. Under exchangeability, conformal regression then constructs a valid $(1 - \alpha)$ coverage interval via the calibration set, and computes the predicted interval according to the quantiles:

$$\hat{C}_{1-\alpha}(x) := \hat{f}(x) \pm (1 - \alpha) \text{ quantile of } \{S(\mathbf{x}_i, y_i)\}_{i \in \mathcal{I}_{\text{cal}}}. \tag{1}$$

However, in non-stationary settings such as time-series forecasting, the exchangeability assumption fails. A common remedy is a sliding calibration window using recent residuals for threshold estimation, resembling local exchangeability (Campbell et al., 2019). While theoretical coverage guarantees are more difficult to establish, these adaptive methods (Xu & Xie, 2021; Gibbs & Candes, 2021; Xu & Xie, 2023b) have been shown to maintain approximate validity under mild assumptions. Formally, at each time $t$, given a stream of observations $\{(\mathbf{x}_i, y_i)\}_{i=t-w+1}^{t}$ with $\mathbf{x}_t \in \mathbb{R}^{d \times w}$, we construct a sliding calibration window of size $w_c$ ($\leq t$), and estimates conformity threshold $\hat{q}_{1-\alpha}^{(w_c, t)}$ from the calibration window and form the interval $\hat{C}_{1-\alpha}^{(w_c, t)}$.

# 3 METHOD

Uncertainty quantification is essential for developing robust and trustworthy machine learning systems. CP offers rigorous finite-sample coverage guarantees, but struggles in non-stationary settings due to static calibration (Gibbs & Candes, 2021). RL enables dynamic exploration and online learning, yet lacks uncertainty estimation and may exhibit overconfident or unstable behavior (Fox et al., 2015). In this work, we propose CORE, a unified framework that integrates CP and RL via feedback-driven calibration to ensure validity and enable uncertainty-aware learning in dynamic environments.

We begin by formulating problem in the standard RL framework, casting conformal regression as a sequential decision-making process. Building on this formulation, we organize the framework around three functional aspects: conformal exploration, where the agent samples quantile-distributed actions to guide exploration under uncertainty; adaptation-aware calibration, which dynamically adjusts conformity scores based on exploration results; and uncertainty-guided rewards, which jointly regulate both exploration and calibration to enhance predictive reliability.

## 3.1 RL FORMULATION

We cast adaptive conformal regression as a reinforcement learning (RL) problem, where the agent learns to construct valid and efficient prediction intervals through interaction with the data-generating environment. Formally, we model this process as a Markov Decision Process (MDP), represented by a 5-tuple $(\mathcal{S}, \mathcal{A}, \mathcal{T}, \mathcal{R}, \beta)$. State $\mathcal{S}$: at time $t$, the state $\mathbf{s}_t$ encodes a temporal feature window $\mathbf{x}_t \in \mathbb{R}^{d \times w}$, where $w$ is the window size. Action $\mathcal{A}$: the agent outputs quantile-based predictions $\mathbf{a}_t$, which specify a prediction interval $\hat{C}_{1-\alpha_t}^{(w_c, t)} \subseteq \mathbb{R}$ for the next response $y_t$. Reward $\mathcal{R}$: feedback is provided through conformity scores computed from recent samples. The reward balances calibration validity (whether $y_t \in \hat{C}_{1-\alpha}^{(w_c, t)}$) with efficiency (interval width). Transition $\mathcal{T}$: the next state $s_{t+1}$ naturally arises from the temporal evolution of the sequence, reflecting potential distribution shifts in the data. The agent thus learns a policy $\pi_\theta : \mathcal{S} \to \mathcal{A}$ that maximizes the expected discounted return $R_t = \mathbb{E}\big[\sum_{k=0}^{\infty} \beta^k r_{t+k}\big]$, where calibration feedback is integrated into the reward signal. We adopt Proximal Policy Optimization (PPO) (Schulman et al., 2017) to enable stable and efficient learning in the continuous action space.

## 3.2 CONFORMAL EXPLORATION

We consider the usual setting of a Markov Decision Process (MDP), in which an agent interacts with the environment $\mathcal{E}$. At each time step $t$, the observation $\mathbf{s}_t$ is defined as a temporal window of the past features, i.e., $\mathbf{s}_t = \mathbf{x}_{t-w+1:t} \in \mathbb{R}^{d \times w}$, where $w$ is the window size and $d$ is the feature dimension.

### 3.2.1 QUANTILE-GUIDED ACTOR-CRITIC

In regression, predicting $y_t$ can be viewed as selecting an action in a continuous space. Instead of a point estimate, we model the predictive process as a distributional policy $\pi_\theta$, which samples quantile-based predictions $\mathbf{a}_t = \{a_t^{(q)}\}_{q \in \mathcal{Q}}$ where $\mathcal{Q} := \{q^{(1)}, \ldots, q^{(K)}\}$ denotes $K$ fixed quantiles. For example, with $\mathcal{Q} = \{0.05, 0.5, 0.95\}$, the actor samples three actions of $5^{\text{th}}$ percentile, median, and $95^{\text{th}}$ percentile, where quantile regression captures local heteroscedasticity and yields adaptive prediction intervals that reflect varying uncertainty (Takeuchi et al., 2006; Romano et al., 2019).

The actor-critic architecture strengthens this formulation: the actor generates quantile predictions to represent uncertainty, while the critic provides value-based feedback to stabilize training and improve

efficiency under noisy targets (Haarnoja et al., 2018). Thus, the task becomes a sequential decision process where the agent refines predictions through exploration and policy updates.

### 3.2.2 TRAINING OBJECTIVES

To enable uncertainty-aware prediction, outputs must not only be diverse but also reflect the underlying distribution structure (Meinshausen & Ridgeway, 2006). We therefore adopt a dual-objective scheme enforcing both quantile correctness and pointwise accuracy.

**Quantile Regression Loss**. Each output $a_t^{(q)}$ is optimized to approximate the true conditional quantile at level $q$ by minimizing the tilted (pinball) loss:

$$\mathcal{L}\text{quantile} = \sum_{q \in \mathcal{Q}} \rho_q \left( y_t - a_t^{(q)} \right) \quad \text{with} \quad \rho_q(u) = u \left( q - \mathbb{I}[u < 0] \right), \tag{2}$$

This asymmetric loss penalizes over-estimation and under-estimation differently, guiding the model to align each quantile prediction with its statistical meaning. As a result, the agent learns to represent the spread and skewness of the conditional outcome distribution.

**Behavior Cloning Loss**. Quantile regression enforces distributional consistency, but it does not explicitly ensure that the central prediction aligns with the ground-truth value. We therefore introduce a behavior cloning loss between the median prediction $a_t^{(0.5)}$ and the observed target $y_t$, anchoring the distribution to reduce variance and accelerate early convergence.

### 3.3 ADAPTIVE-AWARE CALIBRATION

We adopt a trajectory-level calibration strategy that adaptively adjusts the conformal confidence to handle non-stationary dynamics. Unlike a fixed confidence level case $1 - \alpha,$, $\alpha_t$ is updated online based on recent calibration error, maintaining both validity and tightness of prediction intervals.

At each timestep $t$, we compute a conformity score $S(\mathbf{x}_t, y_t) = |\hat{f}(a^{(0.5)}) - y_t|$, and append it to a sliding window buffer $\mathcal{S}_t$ of size $w_c$. To capture temporal dynamics, we use the trajectory of median actions $\mathcal{T}_{t-w_c+1:t} := \{a_\tau\}_{\tau=t-w_c+1}^{t}$ together with their conformity scores, estimate the $(1 - \alpha_t)$ quantile $\hat{q}_{1-\alpha_t}^{(w_c,t)}$, and form prediction intervals. At each timestep $t$, the actor outputs quantile-based actions, with the smallest and largest denoted by $a_t^{(\text{low})} := a_t^{(q^{(1)})}$ and $a_t^{(\text{up})} := a_t^{(q^{(K)})}$, respectively. The prediction interval is therefore given by:

$$\hat{C}_{1-\alpha_t}^{(w_c,t)}(\mathbf{x}_t) = [a_t^{(\text{low})} - \hat{q}_{1-\alpha_t}^{(w_c,t)}, \ a_t^{(\text{up})} + \hat{q}_{1-\alpha_t}^{(w_c,t)}]. \tag{3}$$

Let the calibration index set be $\mathcal{I}_t$, so that $\mathcal{S}_t = \{S(\mathbf{x}_i, y_i) : i \in \mathcal{I}_t\}$. Using the sliding window calibration set, we then estimate the empirical miscoverage and update the confidence level via exponential smoothing under the law:

$$\hat{e}_t = \frac{1}{|\mathcal{I}_t|} \sum_{i \in \mathcal{I}_t} \mathbf{1} \left\{ y_i \notin \hat{C}_{1-\alpha_t}^{(w_c,i)} \right\}, \quad \alpha_{t+1} = \alpha_t + \gamma(\alpha - \hat{e}_t), \tag{4}$$

where $\gamma \in \mathbb{R}^+$ controls the update rate and $\alpha$ is the target miscoverage. Calibrating over recent trajectories ensures that validity reflects cumulative performance rather than isolated steps, providing trajectory-aware uncertainty quantification that adapts to temporal patterns and avoids both under-confident and overly conservative intervals.

### 3.4 UNCERTAINTY REWARDS

To guide the learning of an uncertainty-aware policy, we design a composite reward that encourages predictive accuracy, calibrated coverage, and the construction of compact and informative prediction intervals. The overall reward at time $t$ is defined as $r_t = \lambda_{\text{acc}} \cdot r_{\text{acc}} + \lambda_{\text{len}} \cdot r_{\text{len}} + \lambda_{\text{cov}} \cdot r_{\text{cov}}$, where each term reflects a different aspect of performance under uncertainty.

**Accuracy-driven Signal**. To encourage accurate point predictions, we introduce a reward signal based on the absolute deviation between the predicted median and the true target:

$$r_{\text{acc}} = \sigma \big( 1 - \text{residual}(a_t^{(0.5)}, y_t) \big). \tag{5}$$

**Algorithm 1:** CORE, Adaptive Conformal Regression via Reinforcement Learning

**Input:** Data $\{(\mathbf{x}_t, y_t)\}_{t=1}^T$; PPO policy $\pi_\theta$, critic $V_\psi$, learning rate $\eta$; Significance level $\alpha$, quantile levels $\mathcal{Q} = \{q^{(1)}, \ldots, q^{(K)}\}$; Conformity score function $S(\cdot, \cdot)$, observation window size $w$, calibration window size $w_c$, adjustment rate $\gamma$

**Output:** Calibrated policy $\pi_{\theta*}$ with uncertainty-aware policy; Prediction intervals $\hat{C}_{1-\alpha_t}^{(w_c,t)}(\mathbf{x}_t)$

1 Initialize conformity score buffer $\mathcal{S} \leftarrow \emptyset$, prediction interval buffer $\mathcal{C} \leftarrow \emptyset$, ground-truth buffer $\mathcal{Y} \leftarrow \emptyset$;
2 **for** *each training iteration* **do**
3      Initialize environment $\mathcal{E}$ and receive initial window $\mathbf{x}_{1:w}$;
4      **for** $t = w$ **to** $T$ **do**
5          Observe state $\mathbf{s}_t = \mathbf{x}_{t-w+1:t}$;
6          Draw actions $\{a_t^{(q)}\}_{q \in \mathcal{Q}} \sim \pi_\theta(\mathbf{s}_t)$ and let $\bar{\mathbf{a}}_t = a_t^{(0.5)}$;
7          Update $\mathcal{S} \leftarrow \mathcal{S} \cup \{S(a_t^{0.5}, y_t)\}$;
8          **if** $|\mathcal{S}| > w_c$ **then** Remove oldest score from $\mathcal{S}$ to maintain $|\mathcal{S}| = w_c$;
9          Update buffers $\mathcal{C} \leftarrow \mathcal{C} \cup \{a_t^{0.5}\}$, $\mathcal{Y} \leftarrow \mathcal{Y} \cup \{y_t\}$, then compute error rate $\hat{e}_t$ using $(\mathcal{C}, \mathcal{Y})$;
10          $\alpha_{t+1} \leftarrow \alpha_t + \gamma(\alpha - \hat{e}_t)$;                // adaptive confidence level
11          **if** $|\mathcal{S}| < w_c$ **then** Set $\hat{q}_{1-\alpha_t}^{(w_c,t)} \leftarrow 0$;
12          **else** $\hat{q}_{1-\alpha_t}^{(w_c,t)} \leftarrow \text{Quantile}_{1-\alpha_t}(\mathcal{S})$;
13          $\hat{C}_{1-\alpha_t}^{(w_c,t)}(\mathbf{x}_t) \leftarrow [a_t^{(\text{low})} - \hat{q}_{1-\alpha_t}^{(w_c,t)}, a_t^{(\text{up})} + \hat{q}_{1-\alpha_t}^{(w_c,t)}]$;
14          Compute reward $r_t = \lambda_{\text{acc}} \cdot r_{\text{acc}} + \lambda_{\text{len}} \cdot r_{\text{len}} + \lambda_{\text{cov}} \cdot r_{\text{cov}}$;      // refer to Eq.5,6 and 7
15          Observe $y_t$ and next state $\mathbf{s}_{t+1}$;
16          Store transition $(\mathbf{s}_t, \bar{\mathbf{a}}_t, r_t, \mathbf{s}_{t+1})$ in PPO buffer $\mathcal{T}$;
17      Update Critic $V_\psi$ with trajectory $\mathcal{T}$ using generalized advantage estimation (GAE) loss;
18      Update Actor policy $\pi_\theta$ with trajectory $\mathcal{T}$ using quantile loss $\mathcal{L}_{\text{quantile}}$ and behavior cloning loss $\mathcal{L}_{\text{BC}}$;

The $\ell_1$ residual provides robustness to outliers and naturally soft-clips rewards by bounding gradient growth compared to quadratic loss (Haarnoja et al., 2018) To further stabilize training, we apply a $\sigma(\cdot)$ activation to clip extreme values and maintain a consistent reward scale. The reward design is task-adaptive: residuals are used for regression, while cross-entropy or margin-based terms apply for classification. Specifically, for classification task like anomaly detection, with label $y_t \in \{0, 1\}$ and predicted probability $p_t$, we define $r_{\text{acc}} = 1 - \ell_{\text{BCE}}(y_t, p_t) = 1 - \big(y_t \log p_t + (1 - y_t) \log(1 - y_t)\big)$.

**Compactness-aware Interval Penalty**. To balance coverage rate and interval length, we note that overly wide bands diminish the practical value of uncertainty estimates (Romano et al., 2019). We therefore design the following penalty:

$$r_{\text{len}} = 1 - \frac{\left| \hat{C}_{1-\alpha_t}^{(w_c,t)}(\mathbf{x}_t) \right|}{\tilde{y} + \delta} \quad \text{with} \quad \tilde{y} = \text{Median}(\{y_i\}_{i=t-w_c+1}^t), \tag{6}$$

where the numerator is the interval length and the denominator normalizes it by the recent target magnitude (using the local median as a robust reference); and a small $\delta$ avoids division-by-zero. This discourages intervals excessively wide relative to local scale, while $\alpha_t$ controls under-coverage.

**Coverage-consistency Reward**. To ensure that predictions remain consistent with the calibrated interval, we design a reward encouraging the true response and predicted median to be contained and lie inside the interval, respectively:

$$r_{\text{cov}} = \begin{cases} 1 & \text{if } y_t \in \hat{C} \ \wedge \ \bar{\mathbf{a}}_t \in \hat{C} \\ -\beta & \text{if } y_t \notin \hat{C} \ \wedge \ \bar{\mathbf{a}}_t \in \hat{C} \quad \text{or} \quad y_t \in \hat{C} \ \wedge \ \bar{\mathbf{a}}_t \notin \hat{C}, \\ -1 & \text{if } y_t \notin \hat{C} \ \wedge \ \bar{\mathbf{a}}_t \notin \hat{C} \end{cases} \tag{7}$$

where $\hat{C}$ is the surrogate for $\hat{C}_{1-\alpha}^{w_c,t}(\mathbf{x}_t)$, and $\beta \in [0, 1]$, applying graded penalties depending on whether the true value and/or the predicted median fall outside the interval. The asymmetric factor $\beta$ reflects our preference for penalizing under-coverage more heavily, while being more tolerant to over-coverage. Notably, in practice, we fix $\beta = 1$, and $\lambda_{\text{acc}} = \lambda_{\text{len}} = \lambda_{\text{cov}} = 1.0$ across all datasets, without any dataset-specific tuning. This choice follows the heuristic practice of shaping rewards with task knowledge rather than extensive tuning (Cheng et al., 2021; Gupta et al., 2022). Further implementation details are provided in Appendix B.4.

# 4 THEORETICAL ANALYSIS

## 4.1 DATA ASSUMPTIONS

The theoretical results rely on several mild assumptions on the data-generating process, which ensure that the learning dynamics remain well-behaved over time.

**Weak dependence.** The sequence $(X_t, Y_t)_{t \geq 1}$ is allowed to be dependent across time. To control this dependence, we adopt a $\beta$-mixing assumption as in (Xu & Xie, 2023a): $\beta(k) = \sup_{t \geq 1} \sup_{A \in \mathcal{F}_{1:t}} \sup_{B \in \mathcal{F}_{t+k:\infty}} |\Pr(A \cap B) - \Pr(A)\Pr(B)|$. The *summable-$\beta$* condition $\sum_{k=1}^{\infty} \beta(k) < \infty$ ensures that temporal correlations decay sufficiently fast, yielding concentration bounds comparable to the i.i.d. case while accommodating the dependence structure of RL trajectories.

**Local regularity of residual CDF.** For each $t$, let $F_t(z) = \Pr(\varepsilon_t \leq z)$ denote the conditional CDF of the residuals. We assume that $F_t$ is locally Lipschitz, i.e., there exists a constant $L > 0$ such that for all $u, v \in \mathbb{R}$: $|F_t(u) - F_t(v)| \leq L|u - v|$.

**Bounded drift.** We allow the regression function $f_t(x) = \mathbb{E}[Y_t \mid X_t = x]$ to evolve over time, but require its temporal change to be uniformly bounded: $\xi_t := \sup_{x \in \mathbb{R}^d} |f_{t+1}(x) - f_t(x)|, \quad \sum_t \xi_t < \infty$. We denote by $\xi_{s:t} := \sum_{k=s}^{t} \xi_k$ the cumulative drift over the interval $[s, t]$.

## 4.2 MAIN RESULTS

We establish concentration bounds for the empirical residual distribution, link policy optimization to prediction error decay, and ensure stability via adaptive calibration. Together, these yield a uniform validity and efficiency theorem. A complete proof is provided in Appendix D.

**Lemma 1** (Rio-type DKW inequality). *Let $\tilde{F}$ denote the empirical CDF of the true residuals within $[t - w_c + 1, t]$ over a calibration window of size $w_c$, and let $F$ denote the true CDF. Under the summable-$\beta$ weak dependence condition,*

$$\sup_x |\tilde{F}(x) - F(x)| \leq C_1 (\log w_c / w_c)^{1/3}.$$

**Lemma 2** (Quantile-error transfer). *Let $\delta_T^2 = \frac{1}{T} \sum_{t=1}^{T} (\hat{\varepsilon}_t - \varepsilon_t)^2$. If $F_t$ is L-Lipschitz, then*

$$\sup_x |\hat{F}(x) - \tilde{F}(x)| \leq C_2 \delta_T^{2/3} + 2 \sup_x |\tilde{F}(x) - F(x)|,$$

*where $C_2 := L + 1$, with the additive constant serving as a technical slack.*

To obtain sharper guarantees, it remains to show that these quantities diminish under the policy optimization dynamics of PPO.

**Proposition 1** (Prediction-error decay under PPO). *Suppose the policy is updated by PPO with step sizes satisfying $\sum_k \eta_k = \infty$ and $\sum_k \eta_k^2 < \infty$, then the median prediction error decays as*

$$\delta_T^2 := \frac{1}{T} \sum_{t=1}^{T} \mathbb{E}\left[(Y_t - f_t(X_t))^2\right] = \mathcal{O}_p(T^{-p/2}), \quad \text{for some } p > 0.$$

In addition, the adaptive update of the miscoverage level stabilizes empirical coverage.

**Proposition 2** (Adaptive $\alpha$ stability). *Let $\alpha_{t+1} = \alpha_t + \gamma(\alpha - \hat{\epsilon}_t)$, where $\hat{\epsilon}_t = \mathbf{1}\{Y_t \notin \hat{C}_t\}$. Then the gap between the running level $\alpha_t$ and the true miscoverage $\epsilon_t$ satisfies*

$$|\alpha_t - \epsilon_t| = \mathcal{O}(\gamma^{1/2}).$$

Finally, putting these results together yields the main guarantee.

**Theorem 1** (Master Theorem: validity and efficiency). *Combining Lemmas 1 and 2, Proposition 1, and 2, the* CORE *intervals satisfy:*

*(**Validity**)* *For any horizon $T$,*
$$\sup_{t \leq T} \left|\Pr\{Y_t \in \hat{C}_{1-\alpha}^{(w,t)}\} - (1 - \alpha)\right| = \mathcal{O}\left((\log w_c / w_c)^{1/3} + T^{-p/3} + \gamma^{1/2}\right).$$

*(**Efficiency**)* *Let $C_t^*$ denote the oracle shortest valid interval. Then*
$$\sum_{t=1}^{T} \left(|\hat{C}_t| - |C_t^*|\right) = \tilde{\mathcal{O}}(T^{1-p/3}).$$

### 4.3 DISCUSSION

(1) **Adaptive calibration stabilizes miscoverage.** Proposition 2 shows that adaptively updating $\alpha_t$ bounds its deviation from true miscoverage $e_t$ by $\mathcal{O}(\gamma^{1/2})$, ensuring calibration error remains small and responsive to violations without accumulating long-term bias.

(2) **Prediction error drives coverage accuracy.** Proposition 1 shows that under PPO training with diminishing step sizes, the prediction error $\delta_T^2$ vanishes at a polynomial rate. Lemma 2 links this to quantile error, directly tightening coverage bounds.

(3) **The coverage theorem guarantees validity and efficiency.** The Master Theorem 1 shows that `CORE` achieves near-nominal coverage with intervals approaching the shortest valid level at rate $\tilde{O}(T^{1-p/3})$, reflecting adaptive self-correction under decreasing prediction error and mild drift.

## 5 EXPERIMENT

In this section, we evaluate the predictive performance, reliability, and adaptability of `CORE`. Specifically, we have following experiments: We evaluate `CORE` on standard time-series forecasting datasets to show that it produces valid uncertainty intervals and accurate predictions under distribution shifts. Results on the mean performance are reported in Section 5.2, while the complete results including both mean and variance are provided in Appendix C.1. Section 5.3 analyzes hyperparameter sensitivity with respect to the adjustment rate $\gamma$, calibration window size $w_c$, and quantile set $\mathcal{Q}$, highlighting trade-offs between validity and efficiency. Finally, Section 5.4 extends `CORE` to an anomaly detection variant `CORE-AD`, showing superior AUC and generalization beyond regression.

### 5.1 EXPERIMENTAL SETUP

Table 1: Experimental setup: tasks, datasets, metrics, and baselines. Detailed configurations of hardware and hyperparameters are in Appendix B.4.

| Task | Datasets | Metrics | Baselines |
|------|----------|---------|-----------|
| Uncertainty Quantification | Weather (Zhou et al., 2021) 
 Traffic (Lai et al., 2017) 
 Electricity (Lai et al., 2017) 
 Illness (Lai et al., 2017) 
 ETT (Zhou et al., 2021) | Coverage Rate (CVG) 
 Interval Length (LEN) | SPCI (Xu & Xie, 2023b) 
 ACI (Gibbs & Candes, 2021) 
 EnbPI (Xu & Xie, 2021) |
| Point Prediction | Same as above | Mean Squared Error (MSE) 
 Mean Absolute Error (MAE) | TimeXer (Wang et al., 2024) 
 iTransformer (Liu et al., 2023) 
 DLinear (Zeng et al., 2023) |
| Anomaly Detection | SMD (Xu et al., 2022) 
 SMAP (Hundman et al., 2018) 
 SWaT (Mathur & Tippenhauer, 2016) | Receiver Operating Characteristic (ROC) 
 Area Under the Curve (AUC) | LSTM (Hundman et al., 2018) 
 IForest (Liu et al., 2008) 
 DeepSVDD (Ruff et al., 2018) |

### 5.2 INTERVAL VALIDITY AND PREDICTION ACCURACY

We study eight time-series datasets: Table 2 shows mean results on four representative datasets, while the appendix (Tables B.2, B.3) reports full results with mean and standard deviation across all eight. These confirm `CORE`'s consistently strong performance.

For *uncertainty quantification*, `CORE` consistently outperforms conformal baselines across datasets. As shown in Figure 2 (a), it achieves the highest coverage with narrow prediction intervals. Compared to the best-performing baselines, it reduces LEN by an average of 5.03% without compromising coverage, indicating tighter yet reliable prediction intervals. Figure 2 (b) shows that `CORE` effectively capture temporal shifts and extreme values by selectively adjusting interval width, closely tracking the ground truth while avoiding the uniform widening seen in EnbPI and others.

For *point prediction accuracy*, `CORE` also ranks among the top models in terms of predictive accuracy, attaining the lower MSE and competitive MAE performance. Its advantage becomes more pronounced in long-horizon settings ($T = 336, 720$), where it achieves an average improvement of 3.16% in MSE and 5.16% in MAE, compared to the best-performing baseline method. These results confirm that `CORE` delivers both precise forecasts and calibrated uncertainty under a unified evaluation framework, and remains robust under increased error accumulation and temporal drift.

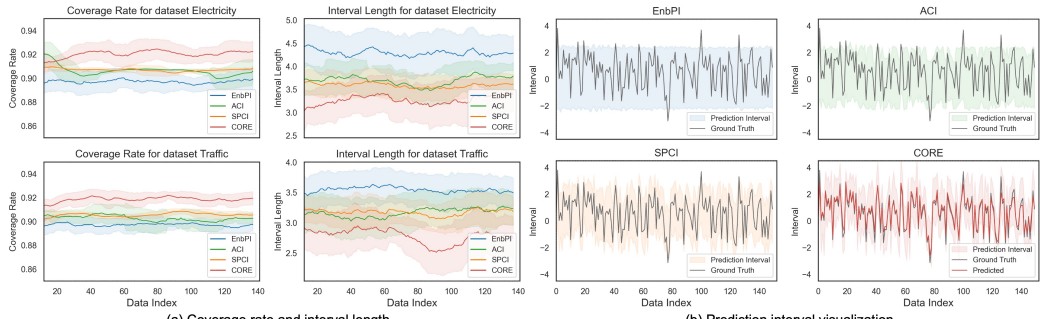

Figure 2: Prediction Interval Comparison Between CORE and Baselines.

## 5.3 HYPERPARAMETER SENSITIVITY

To evaluate the robustness of CORE under non-stationary conditions, we conduct a hyperparameter sensitivity study focusing on three key components: the adjustment rate $\gamma$ for confidence adaptation, the calibration window size $w_c$, and the quantile set $\mathcal{Q}$, whose granularity is controlled by varying the number of quantile levels used in prediction. These parameters govern the dynamic calibration process and directly shape the trade-off between predictive validity and efficiency.

In the experiments, we vary each hyperparameter individually while keeping the others fixed, and evaluate performance using CVG and LEN. As shown in Figure 3, increasing the adjustment rate $\gamma$ accelerates correction of under-coverage but widens intervals. For the calibration window size $w_c$, small windows adapt quickly but produce noisy estimates, leading to unstable coverage. Larger windows stabilize calibration and reduce LEN, but overly large values reduce adaptability to regime shifts. Empirically, $w_c$ between 40 and 60 achieves stable coverage on Electricity dataset. Expanding the quantile set $\mathcal{Q}$ improves calibration initially by capturing more uncertainty structure, but excessive granularity can introduce redundancy and inflate intervals. We observe that varying the number of quantiles does not cause drastic performance changes, suggesting robustness of CORE. Overall, moderate values for all three hyperparameters tend to yield the best balance between validity and efficiency across diverse settings.

Table 2: Performance comparison for uncertainty quantification and prediction accuracy.

| Models | | CORE | | SPCI | | ACI | | EnbPI | | CORE | | TimeXer | | iTransformer | | DLinear | |
|---|---|---|---|---|---|---|---|---|---|---|---|---|---|---|---|---|---|
| Metric | | LEN | CVG | LEN | CVG | LEN | CVG | LEN | CVG | MSE | MAE | MSE | MAE | MSE | MAE | MSE | MAE |
| Weather | 96[1] | 3.635[2] | 0.934 | 4.035[3] | 0.899 | 4.042 | 0.900 | 4.059 | 0.904 | 0.178 | 0.199 | 0.172 | 0.228 | 0.194 | 0.241 | 0.196 | 0.248 |
| | 192 | 4.065 | 0.922 | 4.225 | 0.899 | 4.278 | 0.899 | 4.238 | 0.909 | 0.225 | 0.241 | 0.231 | 0.253 | 0.248 | 0.286 | 0.241 | 0.293 |
| | 336 | 4.744 | 0.920 | 5.221 | 0.901 | 5.437 | 0.901 | 6.842 | 0.884 | 0.272 | 0.261 | 0.277 | 0.316 | 0.292 | 0.346 | 0.278 | 0.325 |
| | 720 | 5.721 | 0.929 | 7.056 | 0.900 | 7.072 | 0.901 | 8.133 | 0.870 | 0.341 | 0.304 | 0.324 | 0.352 | 0.332 | 0.369 | 0.342 | 0.376 |
| Traffic | 96 | 2.868 | 0.919 | 3.159 | 0.895 | 3.182 | 0.897 | 3.184 | 0.899 | 0.410 | 0.507 | 0.412 | 0.456 | 0.402 | 0.452 | 0.421 | 0.478 |
| | 192 | 2.991 | 0.920 | 3.147 | 0.899 | 3.170 | 0.899 | 3.255 | 0.908 | 0.414 | 0.536 | 0.435 | 0.494 | 0.427 | 0.495 | 0.434 | 0.494 |
| | 336 | 3.088 | 0.925 | 3.170 | 0.903 | 3.140 | 0.902 | 3.194 | 0.898 | 0.418 | 0.512 | 0.458 | 0.523 | 0.442 | 0.517 | 0.458 | 0.512 |
| | 720 | 3.037 | 0.919 | 3.152 | 0.899 | 3.159 | 0.898 | 3.621 | 0.870 | 0.427 | 0.523 | 0.482 | 0.554 | 0.453 | 0.525 | 0.478 | 0.539 |
| Electricity | 96 | 3.380 | 0.928 | 3.545 | 0.897 | 3.649 | 0.897 | 4.366 | 0.898 | 0.135 | 0.250 | 0.136 | 0.251 | 0.144 | 0.253 | 0.151 | 0.273 |
| | 192 | 3.457 | 0.916 | 3.558 | 0.904 | 3.768 | 0.903 | 4.534 | 0.895 | 0.153 | 0.261 | 0.160 | 0.263 | 0.167 | 0.267 | 0.159 | 0.275 |
| | 336 | 3.901 | 0.928 | 3.952 | 0.904 | 3.953 | 0.905 | 4.725 | 0.886 | 0.173 | 0.280 | 0.182 | 0.278 | 0.193 | 0.282 | 0.172 | 0.293 |
| | 720 | 4.466 | 0.930 | 4.547 | 0.896 | 4.555 | 0.895 | 5.167 | 0.882 | 0.189 | 0.301 | 0.203 | 0.301 | 0.211 | 0.326 | 0.218 | 0.319 |
| Illness | 24 | 4.003 | 0.908 | 4.140 | 0.887 | 4.364 | 0.908 | 4.738 | 0.908 | 1.749 | 1.160 | 1.482 | 1.034 | 1.757 | 1.123 | 2.231 | 1.383 |
| | 36 | 3.476 | 0.903 | 4.118 | 0.884 | 4.460 | 0.910 | 4.681 | 0.908 | 1.512 | 1.070 | 1.326 | 1.184 | 1.723 | 1.116 | 2.243 | 1.321 |
| | 48 | 4.077 | 0.906 | 4.335 | 0.888 | 4.760 | 0.907 | 4.981 | 0.905 | 1.674 | 1.138 | 1.642 | 1.258 | 1.802 | 1.197 | 2.318 | 1.382 |
| | 60 | 3.916 | 0.901 | 4.409 | 0.886 | 4.805 | 0.891 | 5.658 | 0.893 | 1.585 | 1.108 | 1.569 | 1.245 | 1.852 | 1.342 | 2.412 | 1.435 |
| ETTh1 | 96 | 4.837 | 0.925 | 4.868 | 0.899 | 4.880 | 0.899 | 5.481 | 0.901 | 0.386 | 0.438 | 0.412 | 0.419 | 0.398 | 0.423 | 0.392 | 0.413 |
| | 192 | 4.577 | 0.916 | 4.970 | 0.905 | 5.076 | 0.905 | 6.531 | 0.895 | 0.413 | 0.431 | 0.418 | 0.421 | 0.417 | 0.428 | 0.428 | 0.437 |
| | 336 | 5.083 | 0.913 | 5.294 | 0.904 | 5.307 | 0.905 | 7.224 | 0.909 | 0.414 | 0.465 | 0.429 | 0.438 | 0.433 | 0.445 | 0.437 | 0.469 |
| | 720 | 5.288 | 0.910 | 5.731 | 0.899 | 5.851 | 0.900 | 7.371 | 0.879 | 0.445 | 0.457 | 0.437 | 0.441 | 0.447 | 0.472 | 0.454 | 0.482 |
| ETTh2 | 96 | 4.509 | 0.915 | 5.104 | 0.898 | 5.113 | 0.899 | 5.294 | 0.900 | 0.291 | 0.303 | 0.294 | 0.295 | 0.316 | 0.356 | 0.437 | 0.455 |
| | 192 | 4.860 | 0.923 | 5.112 | 0.896 | 5.126 | 0.896 | 5.366 | 0.901 | 0.296 | 0.323 | 0.323 | 0.338 | 0.362 | 0.382 | 0.484 | 0.492 |
| | 336 | 5.229 | 0.917 | 5.356 | 0.901 | 5.460 | 0.905 | 4.741 | 0.876 | 0.298 | 0.347 | 0.374 | 0.384 | 0.412 | 0.398 | 0.553 | 0.528 |
| | 720 | 5.352 | 0.914 | 5.676 | 0.902 | 5.729 | 0.901 | 9.112 | 0.892 | 0.391 | 0.390 | 0.408 | 0.442 | 0.463 | 0.437 | 0.592 | 0.537 |
| ETTm1 | 96 | 4.425 | 0.901 | 4.697 | 0.899 | 4.700 | 0.899 | 6.437 | 0.901 | 0.327 | 0.332 | 0.328 | 0.351 | 0.336 | 0.357 | 0.343 | 0.368 |
| | 192 | 4.368 | 0.906 | 4.685 | 0.895 | 4.711 | 0.895 | 7.040 | 0.899 | 0.339 | 0.348 | 0.346 | 0.348 | 0.361 | 0.397 | 0.351 | 0.384 |
| | 336 | 5.061 | 0.909 | 4.716 | 0.899 | 4.728 | 0.900 | 8.183 | 0.890 | 0.361 | 0.343 | 0.377 | 0.392 | 0.403 | 0.435 | 0.382 | 0.407 |
| | 720 | 5.131 | 0.911 | 4.812 | 0.901 | 4.815 | 0.901 | 8.402 | 0.892 | 0.411 | 0.394 | 0.439 | 0.453 | 0.450 | 0.466 | 0.442 | 0.451 |
| ETTm2 | 96 | 4.203 | 0.898 | 4.437 | 0.896 | 4.445 | 0.895 | 5.265 | 0.903 | 0.228 | 0.293 | 0.234 | 0.283 | 0.253 | 0.302 | 0.275 | 0.328 |
| | 192 | 4.460 | 0.898 | 4.459 | 0.903 | 4.466 | 0.901 | 6.100 | 0.892 | 0.265 | 0.304 | 0.268 | 0.315 | 0.275 | 0.313 | 0.308 | 0.358 |
| | 336 | 4.598 | 0.906 | 4.653 | 0.903 | 4.665 | 0.902 | 6.937 | 0.902 | 0.287 | 0.343 | 0.297 | 0.342 | 0.311 | 0.349 | 0.314 | 0.372 |
| | 720 | 4.947 | 0.917 | 5.149 | 0.898 | 5.251 | 0.901 | 7.236 | 0.875 | 0.333 | 0.372 | 0.331 | 0.376 | 0.334 | 0.374 | 0.356 | 0.394 |

[1] Prediction lengths are $T \in \{24, 36, 48, 60\}$ for Illness and $T \in \{96, 192, 336, 720\}$ for others.
[2] Bold indicates CORE outperforms all baselines.
[3] Underlined marks the best baseline.

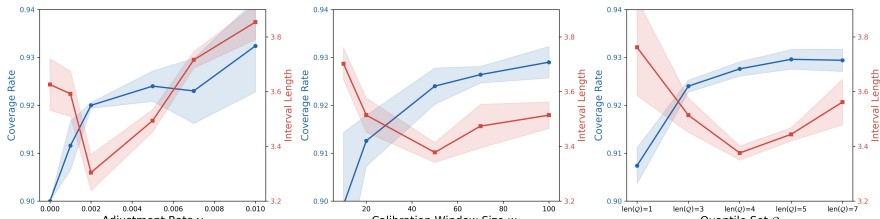

Figure 3: Hyperparameter sensitivity analysis for our proposed method on Electricity dataset

## 5.4 EXTENSION TO ANOMALY DETECTION

To evaluate whether `CORE` can be generalized to other conformal inference tasks beyond regression, we adopt the `ECAD` (Xu & Xie, 2021) setup and introduce a variant, `CORE-AD`, which wraps around the original framework to enable unsupervised anomaly detection. Specifically, it leverages calibrated prediction intervals as decision signals by first computing anomaly scores in a sliding calibration window, where at each $\tau$, $s_\tau = \max\{a_\tau^{(\text{low})} - y_\tau, y_\tau - a_\tau^{(\text{up})}, 0\}$. Then calculate p-value for the current test point as $p_t = \frac{1}{w_c} \sum_{i=1}^{w_c} \mathbf{1}(s_{t-i} \geq s_t)$, which quantifies how atypical the current observation is compared to the calibration set. A point is flagged as anomalous if $p_t \leq \alpha$, ensuring control of the false positive rate. We compare `CORE-AD` against a set of unsupervised and supervised anomaly detectors (Han et al., 2022) on standard time-series anomaly detection datasets (SMD (Xu et al., 2022), SMAP (Hundman et al., 2018), and SWaT (Mathur & Tippenhauer, 2016)). In this task, we evaluate the model's ability to distinguish abnormal from normal time points using **ROC** curves and the area under the curve (**AUC**) as the performance metric. We compare the proposed variant `CORE-AD` against representative unsupervised and semi-supervised baselines, including LSTM (Hundman et al., 2018), IForest (Liu et al., 2008), and DeepSVDD (Ruff et al., 2018).

We visualize the ROC curves of four methods across three datasets to assess anomaly detection performance. As shown in Figure 4, `CORE-AD` exhibits clear separation from competing methods across nearly the entire ROC curve, indicating stronger robustness under different false-positive rate thresholds. These improvements highlight the advantage of leveraging calibrated prediction intervals as decision signals, providing principled uncertainty quantification while preserving high detection accuracy. Overall, the results confirm that `CORE-AD` is not only effective for anomaly detection but also demonstrates its ability to generalize beyond standard regression tasks as a reliable, uncertainty-aware alternative to traditional detectors.

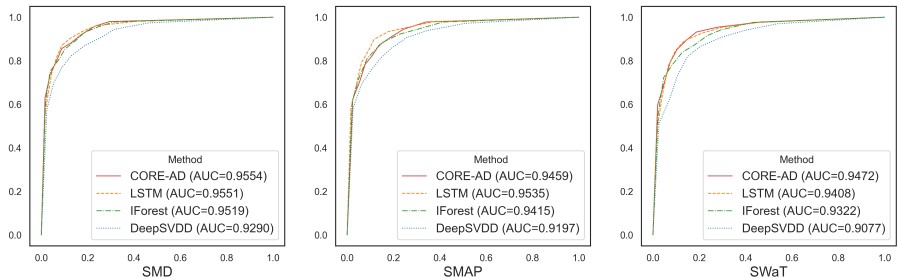

Figure 4: ROC curves (horizontal-axis: false-positive rate; vertical-axis: true-positive rate)

## 6 LIMITATIONS AND CONCLUSION

Our proposed method `CORE` introduces a unified framework that integrates reinforcement learning with conformal prediction through a mutual feedback loop, enabling effective adaptation to distribution shifts. Extensive empirical experiments across eight time-series datasets and state-of-the-art baselines demonstrate `CORE` achieves superior calibrated coverage, interval efficiency, and predictive accuracy. In addition, our theoretical analysis establishes formal guarantees of validity and efficiency under distribution shifts. Despite these advantages, `CORE` also faces several limitations that warrant further study: (i) Cold-Start Instability, where early calibration on limited data leads to unstable rewards and unreliable uncertainty estimates; and (ii) Application Sensitivity, since practical deployment may depend critically on calibration reliability, requiring additional safeguards for system stability. Addressing these challenges represents an important direction for future research.

## REPRODUCIBILITY STATEMENT

We have taken concrete steps to ensure the reproducibility of our work. The complete source code is provided in the supplementary materials. Detailed implementation settings, including model architectures, hyperparameters, and training procedures, are documented in Appendix B.4, while the full proofs of our theoretical results are presented in Appendix D. Together, these materials allow independent researchers to replicate both our empirical results and theoretical analysis.

## THE USE OF LLM

The authors confirm that large language models (LLMs), such as ChatGPT, were only used during the writing stage of this paper for editorial assistance, including grammar correction, spelling checks, and refinement of phrasing. No LLMs were employed in the design of the methodology, execution of experiments, analysis of data, or interpretation of results. All scientific and technical content was solely conceived and produced by the authors.

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

# Appendices

## A  RELATED WORK

### A.1  CONFORMAL PREDICTION

Conformal Prediction (CP) is a distribution-free framework for constructing prediction intervals with formal coverage guarantees under minimal assumptions Romano et al. (2019); Shafer & Vovk (2008); Xu & Xie (2021). By leveraging a nonconformity score function and calibration on held-out data Romano et al. (2019), CP methods can produce reliable uncertainty estimates. Over time, CP has evolved from inductive variants to stratified methods like Mondrian CP, and further to adaptive extensions for dynamic, non-stationary environments.

Inductive CP (ICP) forms prediction intervals by splitting data into training and calibration sets Vovk et al. (2005); Lei et al. (2018). Extensions including importance-weighted calibration Tibshirani et al. (2019); Cauchois et al. (2020), ensemble resampling Gupta et al. (2021); Kim et al. (2020), and quantile-based regression Romano et al. (2019); Triebe et al. (2021) improve its robustness. Mondrian CP (MCP) Boström et al. (2021) further improves calibration by conditioning on predefined categories (e.g., class labels or tree leaves), providing group-conditional guarantees Boström et al. (2021); Toccaceli & Gammerman (2019). However, both ICP and MCP rely on fixed calibration sets and assume data exchangeability Xu & Xie (2023b), making them prone to miscalibration and poor adaptability in dynamic scenarios.

Unlike ICP or MCP, online and adaptive CP methods dynamically update prediction intervals using recent residuals to adapt Xu & Xie (2021); Gibbs & Candes (2021). Techniques such as sliding-window residual regression Xu & Xie (2023b), importance weighting Barber et al. (2022), and adaptive significance tuning Gibbs & Candes (2021); Zaffran et al. (2022) help enhance calibration. Among them, EnbPI Xu & Xie (2021) and SPCI Xu & Xie (2023b) maintain validity and efficient intervals under temporal dependence and distribution shifts commonly observed in time-series scenarios. Our work continues focusing on adaptive CP with the goal of providing valid coverage guarantees under non-exchangeable data.

### A.2  CONFORMAL REINFORCEMENT LEARNING

Reinforcement learning (RL) optimizes decision-making through interaction with environments and is widely used to learn policies under uncertainty and delayed feedback. Its ability to model dynamics and explore makes it well-suited for adaptive systems in non-stationary settings.

Despite its strengths, RL often suffers from training instability, policy divergence, and sensitivity to distributional shifts. To mitigate these limitations, recent work has explored integrating CP into RL. For instance, PlanCP Sun et al. (2023) employs CP for calibrated offline planning, while SoNIC Yao et al. (2024) and Egocentric CP Shin et al. (2025) improve inference and safety in navigation. In multi-agent and safety-critical scenarios, CP-based confidence sets and safety filters Gupta et al. (2023); Strawn et al. (2023) support risk-aware coordination. Despite their effectiveness, these methods typically treat CP as an external or post-hoc module, limiting the impact of uncertainty feedback on policy learning.

It is worth noting that distribution shift remains a challenge in RL. Although meta-RL Nagabandi et al. (2018); Xu et al. (2018) and transfer learning Taylor & Stone (2009); Zhu et al. (2023) improve adaptation through experience sharing, they often lack formal uncertainty quantification. To address this gap, we propose integrating CP directly into RL learning process, which enables online detection of unreliable predictions with formal coverage guarantees, thus enhancing the safety and robustness of policy learning under distributional shifts. Unlike prior approaches, our goal is to promote **mutual adaptation**, which not only improves policy reliability in non-stationary environments, but also dynamically refines the CP calibration process based on the agent's evolving interaction history.

# B EXPERIMENTAL SETUP

## B.1 DATASETS

To assess the effectiveness of our proposed CORE in long-term time-series forecasting, we conduct experiments on eight diverse, real-world datasets that are widely used across various works:

- Weather Zhou et al. (2021): Collected at 10-minute intervals in 2020, this dataset includes 21 meteorological variables, with Wet Bulb temperature as the prediction target and the others as auxiliary features.

- Traffic Lai et al. (2017): This dataset contains hourly occupancy rates from 862 freeway sensors in the San Francisco Bay Area, using the last sensor's data as the target and the others as contextual inputs.

- Electricity Lai et al. (2017): This dataset contains hourly electricity consumption data from 321 residential clients (2012–2014). Typically, the last client's usage is the prediction target, with the others as exogenous inputs.

- Illness Lai et al. (2017): This dataset contains weekly influenza-like illness rates across multiple U.S. regions, with the goal of forecasting one region's trend using its own and others' historical data as inputs.

- ETT Zhou et al. (2021): The ETT dataset includes four subsets, ETTh1/ETTh2 (hourly) and ETTm1/ETTm2 (15-minute), with oil temperature as the target and six power metrics as features, spanning July 2016 to July 2018 across two Chinese counties.

To assess the ablility of variant model, CORE-AD, we conduct we experiments on some standard time-series anomaly detection datasets:"

- SMAP Hundman et al. (2018) Soil moisture active passive dataset is collected from NASA's Earth observation satellite, containing 25-dimensional data used for anomaly detection in spacecraft systems.

- SMD Xu et al. (2022) Server machine dataset is collected from a large internet company, consisting of 38-dimensional sensor readings over 5 weeks.

- SWaT Mathur & Tippenhauer (2016) This dataset was collected from a real-world secure water treatment testbed, containing 51 tagged sensor and actuator signals recorded at 1-second intervals over 11 days.

## B.2 METRICS

For uncertain evaluation, we adopt two standard metrics.

- **Coverage Rate (CVG)**: Measures the proportion of true values covered by the prediction intervals.

$$\text{CVG} = \frac{1}{|D_{\text{test}}|} \sum_{t \in D_{\text{test}}} \mathbb{I}[y_t \in \hat{C}_{1-\alpha}^{(w,t)}], \tag{B.8}$$

where $D_{\text{test}}$ represents the set of test samples, $y_t$ is the ground-truth value, $\hat{C}_{1-\alpha}^{(w,t)}$ is the prediction interval, and $\mathbb{I}[\cdot]$ is the indicator function that returns 1 if the condition holds and 0 otherwise

- **Interval Length (LEN)**: Measures the average length of the prediction intervals.

$$\text{LEN} = \frac{1}{|D_{\text{test}}|} \sum_{t \in D_{\text{test}}} (\hat{u}_t - \hat{l}_t), \tag{B.9}$$

where $\hat{u}_t$ and $\hat{l}_t$ are the upper and lower bounds of the interval for instance $t$.

For accuracy, we report **Mean Squared Error (MSE)** and **Mean Absolute Error (MAE)**, which measure the average squared and absolute deviation between predicted and ground-truth values, respectively.

For anomaly detection task, we evaluate the model's ability to distinguish abnormal from normal time points using receiver operating characteristic(**ROC**) curves, which plot the true positive rate against the false positive rate at varying thresholds. The area under the curve (**AUC**) summarizes this trade-off, with higher values indicating better overall discrimination performance.

- **True Positive Rate (TPR)**: Measures the proportion of actual anomalies that are correctly detected by the model.

$$\text{TPR} = \frac{\text{TP}}{\text{TP} + \text{FN}}, \tag{B.10}$$

  where TP is the number of correctly identified anomalies, and FN is the number of missed anomalies.

- **False Positive Rate (FPR)**: Measures the proportion of normal points that are incorrectly classified as anomalies.

$$\text{FPR} = \frac{\text{FP}}{\text{FP} + \text{TN}}, \tag{B.11}$$

  where FP is the number of normal points incorrectly flagged as anomalies, and TN is the number of correctly identified normal points.

By integrating the ROC curve, AUROC summarizes the trade-off between TPR and FPR across all thresholds, where a higher AUROC value (closer to 1) indicates better overall detection performance.

## B.3 BASELINES

We evaluate the following baselines, categorized by tasks:

- **uncertainty quantification**:
  - SPCI Xu & Xie (2023b) constructs prediction intervals by modeling the conditional quantile of residuals, explicitly accounting for temporal dependence without relying on bootstrap ensembles.
  - EnbPI Xu & Xie (2021) constructs prediction intervals from bootstrap ensembles, achieving approximate coverage without data exchangeability or retraining.
  - ACI Gibbs & Candes (2021) adapts conformal inference to distribution shifts by recalibrating prediction sets over time to maintain target coverage under non-stationary data.
- **point prediction**:
  - TimeXer Wang et al. (2024) is a Transformer-based model that enhances target prediction by selectively attending to exogenous inputs through structured self-attention and cross-variable integration mechanisms.
  - iTransformer Liu et al. (2023) applies standard Transformer components on transposed inputs to better capture inter-variable dependencies without architectural modification.
  - DLinear Zeng et al. (2023) models trend and residual components using simple linear projections, achieving strong forecasting performance with minimal model complexity.
- **anomaly detection**:
  - DeepSVDD Ruff et al. (2018) trains a neural network with an anomaly detection-specific objective, aiming to enclose normal data within a minimal-volume hypersphere in the feature space.
  - IForest Liu et al. (2008) is a tree-based algorithm that detects anomalies by recursively partitioning the feature space and measuring the path length needed to isolate each point.
  - LSTM Hundman et al. (2018) is an unsupervised method, which leverages the long short-term memory recurrent neural networks to achieve prediction performance while maintaining interpretability.

## B.4 IMPLEMENTATION DETAILS

We implement our method in PyTorch and train all models on a single NVIDIA A5000 GPU with 24GB memory. All code is available in the supplementary files. Each training sample is organized into batches with shape $(B, W, D)$, where $B = 128$ denotes the batch size, $W = 50$ is the input temporal

Table B.2: Performance comparison on eight time-series datasets for uncertainty quantification. Prediction lengths are $T \in \{24, 36, 48, 60\}$ for Illness and $T \in \{96, 192, 336, 720\}$ for others. Results are reported as mean ± range over 5 runs. Bold indicates CORE outperforms all baselines; underlined marks the best baseline.

| Models | | CORE | | SPCI Xu & Xie (2023b) | | ACI Gibbs & Candes (2021) | | EnbPI Xu & Xie (2021) | |
|---|---|---|---|---|---|---|---|---|---|
| Metric | | LEN | CVG | LEN | CVG | LEN | CVG | LEN | CVG |
| Weather | 96 | **3.635 ± 1.051** | **0.934 ± 0.012** | 4.035 ± 0.650 | 0.899 ± 0.002 | 4.042 ± 0.286 | 0.900 ± 0.002 | 4.059 ± 0.142 | 0.904 ± 0.001 |
| | 192 | **4.065 ± 1.115** | **0.922 ± 0.016** | 4.225 ± 0.692 | 0.899 ± 0.001 | 4.278 ± 0.208 | 0.899 ± 0.000 | 4.238 ± 0.109 | 0.909 ± 0.001 |
| | 336 | **4.744 ± 0.785** | **0.920 ± 0.033** | 5.221 ± 0.558 | 0.901 ± 0.001 | 5.437 ± 0.901 | 0.901 ± 0.000 | 6.842 ± 0.106 | 0.884 ± 0.009 |
| | 720 | **5.721 ± 1.321** | **0.929 ± 0.032** | 7.056 ± 0.657 | 0.900 ± 0.003 | 7.072 ± 0.729 | 0.901 ± 0.003 | 8.133 ± 0.278 | 0.870 ± 0.012 |
| Traffic | 96 | **2.868 ± 0.370** | **0.919 ± 0.050** | 3.159 ± 0.352 | 0.895 ± 0.033 | 3.182 ± 0.052 | 0.897 ± 0.002 | 3.184 ± 0.141 | 0.899 ± 0.001 |
| | 192 | **2.991 ± 0.510** | **0.920 ± 0.056** | 3.147 ± 0.238 | 0.899 ± 0.002 | 3.170 ± 0.070 | 0.899 ± 0.002 | 3.255 ± 0.078 | 0.908 ± 0.000 |
| | 336 | **3.088 ± 0.505** | **0.925 ± 0.075** | 3.170 ± 0.252 | 0.903 ± 0.002 | 3.140 ± 0.006 | 0.902 ± 0.001 | 3.194 ± 0.125 | 0.898 ± 0.003 |
| | 720 | **3.037 ± 0.727** | **0.919 ± 0.050** | 3.152 ± 0.331 | 0.899 ± 0.003 | 3.159 ± 0.052 | 0.898 ± 0.003 | 3.621 ± 0.032 | 0.870 ± 0.004 |
| Electricity | 96 | **3.380 ± 0.276** | **0.928 ± 0.041** | 3.545 ± 0.273 | 0.897 ± 0.003 | 3.649 ± 0.132 | 0.897 ± 0.004 | 4.366 ± 0.105 | 0.898 ± 0.001 |
| | 192 | **3.457 ± 0.982** | **0.916 ± 0.038** | 3.558 ± 0.124 | 0.904 ± 0.002 | 3.768 ± 0.129 | 0.903 ± 0.001 | 4.534 ± 0.140 | 0.895 ± 0.001 |
| | 336 | **3.901 ± 0.725** | **0.928 ± 0.017** | 3.952 ± 0.221 | 0.904 ± 0.001 | 3.953 ± 0.311 | 0.905 ± 0.003 | 4.725 ± 0.163 | 0.886 ± 0.004 |
| | 720 | **4.466 ± 0.797** | **0.930 ± 0.023** | 4.547 ± 0.341 | 0.896 ± 0.003 | 4.555 ± 0.246 | 0.895 ± 0.004 | 5.167 ± 0.213 | 0.882 ± 0.006 |
| Illness | 24 | **4.003 ± 0.848** | **0.908 ± 0.006** | 4.140 ± 0.132 | 0.887 ± 0.003 | 4.364 ± 0.192 | 0.908 ± 0.006 | 4.738 ± 0.121 | 0.908 ± 0.004 |
| | 36 | **3.476 ± 0.744** | 0.903 ± 0.013 | 4.118 ± 0.044 | 0.884 ± 0.003 | 4.460 ± 0.112 | 0.910 ± 0.004 | 4.681 ± 0.159 | 0.908 ± 0.004 |
| | 48 | **4.077 ± 0.627** | 0.906 ± 0.081 | 4.335 ± 0.148 | 0.888 ± 0.004 | 4.760 ± 0.233 | 0.907 ± 0.006 | 4.981 ± 0.087 | 0.905 ± 0.005 |
| | 60 | **3.916 ± 1.483** | **0.901 ± 0.125** | 4.409 ± 0.163 | 0.886 ± 0.005 | 4.805 ± 0.144 | 0.891 ± 0.007 | 5.658 ± 0.160 | 0.893 ± 0.004 |
| ETTh1 | 96 | **4.837 ± 0.373** | **0.925 ± 0.036** | 4.868 ± 0.192 | 0.899 ± 0.001 | 4.880 ± 0.221 | 0.899 ± 0.001 | 5.481 ± 0.048 | 0.901 ± 0.001 |
| | 192 | **4.577 ± 0.534** | **0.916 ± 0.038** | 4.970 ± 0.132 | 0.905 ± 0.001 | 5.076 ± 0.267 | 0.905 ± 0.001 | 6.531 ± 0.147 | 0.895 ± 0.001 |
| | 336 | **5.083 ± 0.367** | **0.913 ± 0.040** | 5.294 ± 0.181 | 0.904 ± 0.001 | 5.307 ± 0.111 | 0.905 ± 0.001 | 7.224 ± 0.081 | 0.909 ± 0.002 |
| | 720 | **5.288 ± 0.848** | **0.910 ± 0.043** | 5.731 ± 0.112 | 0.899 ± 0.001 | 5.851 ± 0.212 | 0.900 ± 0.001 | 7.371 ± 0.194 | 0.879 ± 0.003 |
| ETTh2 | 96 | **4.509 ± 0.536** | **0.915 ± 0.012** | 5.104 ± 0.236 | 0.898 ± 0.001 | 5.113 ± 0.243 | 0.899 ± 0.001 | 5.294 ± 0.229 | 0.900 ± 0.000 |
| | 192 | **4.860 ± 0.405** | **0.923 ± 0.015** | 5.112 ± 0.232 | 0.896 ± 0.001 | 5.126 ± 0.357 | 0.896 ± 0.001 | 5.366 ± 0.179 | 0.901 ± 0.001 |
| | 336 | **5.229 ± 0.566** | **0.917 ± 0.044** | 5.356 ± 0.322 | 0.901 ± 0.001 | 5.460 ± 0.281 | 0.905 ± 0.001 | 4.741 ± 0.155 | 0.876 ± 0.003 |
| | 720 | **5.352 ± 1.025** | **0.914 ± 0.024** | 5.676 ± 0.240 | 0.902 ± 0.001 | 5.729 ± 0.152 | 0.901 ± 0.001 | 9.112 ± 0.215 | 0.892 ± 0.000 |
| ETTm1 | 96 | **4.425 ± 0.457** | **0.901 ± 0.049** | 4.697 ± 0.212 | 0.899 ± 0.000 | 4.700 ± 0.182 | 0.899 ± 0.000 | 6.437 ± 0.146 | 0.901 ± 0.001 |
| | 192 | **4.368 ± 1.038** | **0.906 ± 0.031** | 4.685 ± 0.216 | 0.895 ± 0.003 | 4.711 ± 0.276 | 0.895 ± 0.003 | 7.040 ± 0.130 | 0.899 ± 0.001 |
| | 336 | 5.161 ± 0.739 | **0.909 ± 0.018** | 4.716 ± 0.137 | 0.899 ± 0.001 | 4.728 ± 0.135 | 0.900 ± 0.000 | 8.183 ± 0.162 | 0.890 ± 0.003 |
| | 720 | 5.131 ± 0.967 | **0.911 ± 0.012** | 4.812 ± 0.140 | 0.901 ± 0.002 | 4.815 ± 0.262 | 0.901 ± 0.001 | 8.402 ± 0.178 | 0.892 ± 0.002 |
| ETTm2 | 96 | **4.203 ± 0.706** | 0.898 ± 0.023 | 4.437 ± 0.157 | 0.896 ± 0.003 | 4.445 ± 0.178 | 0.895 ± 0.003 | 5.265 ± 0.150 | 0.903 ± 0.002 |
| | 192 | **4.460 ± 0.325** | 0.898 ± 0.028 | 4.459 ± 0.136 | 0.903 ± 0.002 | 4.466 ± 0.271 | 0.901 ± 0.000 | 6.100 ± 0.233 | 0.892 ± 0.003 |
| | 336 | **4.598 ± 0.782** | **0.906 ± 0.031** | 4.653 ± 0.135 | 0.903 ± 0.000 | 4.665 ± 0.283 | 0.902 ± 0.000 | 6.937 ± 0.262 | 0.902 ± 0.000 |
| | 720 | **4.947 ± 1.028** | **0.917 ± 0.021** | 5.149 ± 0.210 | 0.898 ± 0.000 | 5.251 ± 0.361 | 0.901 ± 0.001 | 7.236 ± 0.132 | 0.875 ± 0.004 |

window length, and $D$ is the feature dimension, which varies across datasets. The prediction window length $L$ is set to $\{24, 36, 48, 60\}$ for the Illness, and $\{96, 192, 336, 720\}$ for all other datasets. To stabilize training, we adopt a single-step prediction setting, and perform multi-step forecasting during inference to generate outputs of length $l \in L$.

Our method maintains a buffer of conformity scores of fixed size $w_c = 50$ and updates the prediction interval thresholds online at each step using these scores. The adjustment rate $\gamma$ for confidence level $\alpha_t$ is set to $0.002$, and the initial significance level $\alpha$ is fixed at $0.1$. We select $K = 5$ actions from the quantile set $Q = \{q^{(1)}, \ldots, q^{(K)}\}$ to estimate the median actions. The policy is trained using a PPO backbone with separate actor and critic networks, both implemented with Transformer-based architectures. Each network is configured with a depth of 4 layers, 16 attention heads, and a group size of 4. The encoder and decoder modules are flexible and can also be instantiated with alternative architectures such as MLPs.

For optimization, we use the two separate Adam optimizers with a learning rate of $5 \times 10^{-5}$, applied to the policy and value networks respectively. All experiments are run for 1000 training steps per episode, and the model is evaluated every 50 steps using average return and empirical coverage metrics.

# C  EXPERIMENTAL RESULTS

## C.1  EXTENDED EVALUATION OF INTERVAL VALIDITY AND PREDICTION ACCURACY

The experimental results for both uncertainty quantification and prediction accuracy are shown in Table B.2 and Table B.3, respectively, with all values reported as mean ± standard deviation over 5 independent runs.

Across all eight datasets and multiple prediction lengths, CORE consistently achieves high empirical coverage (CVG) while maintaining competitive or shorter interval lengths (LEN) in nearly all

Table B.3: Performance comparison on eight time-series datasets for prediction accuracy. Results are reported as mean ± range over 5 runs. Bold indicates CORE outperforms all baselines; underlined marks the best baseline.

| Models | | CORE | | TimeXer Wang et al. (2024) | | iTransformer Liu et al. (2023) | | DLinear Zeng et al. (2023) | |
|---|---|---|---|---|---|---|---|---|---|
| Metric | | MSE | MAE | MSE | MAE | MSE | MAE | MSE | MAE |
| Weather | 96 | 0.178 ± 0.023 | **0.199 ± 0.020** | 0.172 ± 0.012 | 0.228 ± 0.043 | 0.194 ± 0.012 | 0.241 ± 0.022 | 0.196 ± 0.033 | 0.248 ± 0.019 |
| | 192 | **0.225 ± 0.087** | **0.241 ± 0.036** | 0.231 ± 0.017 | 0.253 ± 0.018 | 0.248 ± 0.019 | 0.286 ± 0.015 | 0.241 ± 0.017 | 0.293 ± 0.034 |
| | 336 | **0.272 ± 0.095** | **0.261 ± 0.046** | 0.277 ± 0.027 | 0.316 ± 0.026 | 0.292 ± 0.029 | 0.346 ± 0.036 | 0.278 ± 0.024 | 0.325 ± 0.027 |
| | 720 | 0.341 ± 0.116 | **0.304 ± 0.049** | 0.324 ± 0.045 | 0.352 ± 0.02 | 0.332 ± 0.019 | 0.369 ± 0.028 | 0.342 ± 0.020 | 0.376 ± 0.026 |
| Traffic | 96 | 0.410 ± 0.013 | 0.507 ± 0.018 | 0.412 ± 0.034 | 0.456 ± 0.013 | 0.402 ± 0.015 | 0.452 ± 0.049 | 0.421 ± 0.024 | 0.478 ± 0.011 |
| | 192 | **0.414 ± 0.011** | 0.536 ± 0.009 | 0.435 ± 0.016 | 0.494 ± 0.018 | 0.427 ± 0.025 | 0.495 ± 0.017 | 0.434 ± 0.034 | 0.494 ± 0.020 |
| | 336 | **0.418 ± 0.055** | **0.512 ± 0.038** | 0.458 ± 0.020 | 0.523 ± 0.045 | 0.442 ± 0.021 | 0.517 ± 0.019 | 0.458 ± 0.036 | 0.512 ± 0.023 |
| | 720 | **0.427 ± 0.072** | **0.523 ± 0.049** | 0.482± 0.027 | 0.554± 0.030 | 0.453 ± 0.019 | 0.525± 0.066 | 0.478± 0.026 | 0.539± 0.012 |
| Electricity | 96 | **0.135 ± 0.006** | **0.250 ± 0.031** | 0.136 ± 0.018 | 0.251 ± 0.009 | 0.144 ± 0.009 | 0.253 ± 0.015 | 0.151 ± 0.019 | 0.273 ± 0.009 |
| | 192 | **0.153 ± 0.007** | **0.261 ± 0.025** | 0.160 ± 0.012 | 0.263 ± 0.020 | 0.167 ± 0.013 | 0.267 ± 0.013 | 0.159 ± 0.010 | 0.275 ± 0.028 |
| | 336 | **0.173 ± 0.015** | 0.280 ± 0.035 | 0.182 ± 0.017 | 0.278 ± 0.016 | 0.193 ± 0.011 | 0.282 ± 0.017 | 0.172 ± 0.02 | 0.293 ± 0.018 |
| | 720 | **0.189 ± 0.043** | **0.301 ± 0.031** | 0.203 ± 0.018 | 0.301 ± 0.019 | 0.211 ± 0.018 | 0.326 ± 0.016 | 0.218 ± 0.019 | 0.319 ± 0.023 |
| Illness | 96 | 1.749 ± 0.063 | 1.160 ± 0.023 | 1.482 ± 0.064 | 1.034 ± 0.082 | 1.757 ± 0.117 | 1.123 ± 0.064 | 2.231 ± 0.195 | 1.383 ± 0.046 |
| | 192 | 1.512 ± 0.123 | **1.070 ± 0.110** | 1.326 ± 0.081 | 1.184 ± 0.053 | 1.723 ± 0.022 | 1.116 ± 0.127 | 2.243 ± 0.068 | 1.321 ± 0.040 |
| | 336 | 1.674 ± 0.058 | **1.138 ± 0.032** | 1.642 ± 0.045 | 1.258 ± 0.057 | 1.802 ± 0.049 | 1.197 ± 0.069 | 2.318 ± 0.054 | 1.382 ± 0.024 |
| | 720 | 1.585 ± 0.147 | **1.108 ± 0.037** | 1.569 ± 0.051 | 1.245 ± 0.062 | 1.852 ± 0.027 | 1.342 ± 0.011 | 2.412 ± 0.077 | 1.435 ± 0.049 |
| ETTh1 | 96 | **0.386 ± 0.009** | 0.438 ± 0.025 | 0.412 ± 0.011 | 0.419 ± 0.009 | 0.398 ± 0.011 | 0.423 ± 0.020 | 0.392 ± 0.018 | 0.413 ± 0.011 |
| | 192 | **0.413 ± 0.044** | 0.431 ± 0.030 | 0.418 ± 0.019 | 0.421 ± 0.013 | 0.417 ± 0.013 | 0.428 ± 0.010 | 0.428 ± 0.011 | 0.437 ± 0.021 |
| | 336 | **0.414 ± 0.057** | 0.465 ± 0.053 | 0.429 ± 0.016 | 0.438 ± 0.010 | 0.433 ± 0.012 | 0.445 ± 0.020 | 0.437 ± 0.016 | 0.469 ± 0.016 |
| | 720 | 0.445 ± 0.058 | **0.457 ± 0.038** | 0.437 ± 0.014 | 0.441 ± 0.015 | 0.447 ± 0.020 | 0.472 ± 0.014 | 0.454 ± 0.025 | 0.482 ± 0.013 |
| ETTh2 | 96 | **0.291 ± 0.047** | 0.303 ± 0.028 | 0.294 ± 0.015 | 0.295 ± 0.007 | 0.316 ± 0.016 | 0.356 ± 0.015 | 0.437 ± 0.014 | 0.455 ± 0.012 |
| | 192 | **0.296 ± 0.014** | **0.323 ± 0.014** | 0.323 ± 0.009 | 0.338 ± 0.008 | 0.362 ± 0.010 | 0.382 ± 0.030 | 0.484 ± 0.017 | 0.492 ± 0.016 |
| | 336 | **0.298 ± 0.025** | **0.347 ± 0.048** | 0.374 ± 0.013 | 0.384 ± 0.009 | 0.412 ± 0.014 | 0.398 ± 0.013 | 0.553 ± 0.030 | 0.528 ± 0.014 |
| | 720 | **0.391 ± 0.122** | **0.390 ± 0.039** | 0.408 ± 0.009 | 0.442 ± 0.014 | 0.463 ± 0.026 | 0.437 ± 0.014 | 0.592 ± 0.012 | 0.537 ± 0.008 |
| ETTm1 | 96 | **0.327 ± 0.049** | **0.332 ± 0.067** | 0.328 ± 0.014 | 0.351 ± 0.011 | 0.336 ± 0.020 | 0.357 ± 0.018 | 0.343 ± 0.015 | 0.368 ± 0.026 |
| | 192 | **0.339 ± 0.034** | **0.348 ± 0.014** | 0.346 ± 0.016 | 0.348 ± 0.023 | 0.361 ± 0.014 | 0.397 ± 0.010 | 0.351 ± 0.021 | 0.384 ± 0.012 |
| | 336 | **0.361 ± 0.051** | **0.343 ± 0.061** | 0.377 ± 0.025 | 0.392 ± 0.020 | 0.403 ± 0.015 | 0.435 ± 0.010 | 0.382 ± 0.034 | 0.407 ± 0.030 |
| | 720 | **0.411 ± 0.074** | **0.394 ± 0.053** | 0.439 ± 0.022 | 0.453 ± 0.014 | 0.450 ± 0.033 | 0.466 ± 0.015 | 0.442 ± 0.026 | 0.451 ± 0.024 |
| ETTm2 | 96 | **0.228 ± 0.062** | 0.293 ± 0.054 | 0.234 ± 0.027 | 0.283 ± 0.013 | 0.253 ± 0.022 | 0.302 ± 0.029 | 0.275 ± 0.014 | 0.328 ± 0.014 |
| | 192 | **0.265 ± 0.039** | **0.304 ± 0.056** | 0.268 ± 0.016 | 0.315 ± 0.028 | 0.275 ± 0.026 | 0.313 ± 0.013 | 0.308 ± 0.023 | 0.358 ± 0.031 |
| | 336 | **0.287 ± 0.027** | 0.343 ± 0.062 | 0.297 ± 0.035 | 0.342 ± 0.014 | 0.311 ± 0.024 | 0.349 ± 0.011 | 0.314 ± 0.018 | 0.372 ± 0.027 |
| | 720 | 0.333 ± 0.055 | **0.372 ± 0.131** | 0.331 ± 0.020 | 0.376 ± 0.029 | 0.334 ± 0.017 | 0.374 ± 0.02 | 0.356 ± 0.030 | 0.394 ± 0.018 |

settings. Compared to ACI and EnbPI, which often struggle to meet the nominal coverage level of 0.9, CORE produces more stable and reliable interval estimates. Visualization of the prediction intervals with ground-truth values further shows that our method effectively adapts to data fluctuations and dynamically adjusts interval widths as needed. These findings confirm the robustness of our uncertainty-aware design across diverse temporal dynamics and forecast horizons.

## D  THEORETICAL PROOF

We recall the Theorem 1 here.

**Theorem 1** (Master Theorem). *The* CORE *intervals satisfy:*

*(**Validity**) For any horizon $T$,*

$$\sup_{t \leq T} \left| \Pr\{Y_t \in \hat{C}_{1-\alpha}^{(w_c,t)}\} - (1-\alpha) \right| = \mathcal{O}\left( (\log w_c/w_c)^{1/3} + T^{-p/3} + \gamma^{1/2} \right).$$

*(**Efficiency**) Let $C_t^*$ denote the oracle shortest valid interval. Then*

$$\sum_{t=1}^{T} \left( |\hat{C}_t| - |C_t^*| \right) = \tilde{\mathcal{O}}(T^{1-p/3}).$$

We first establish an auxiliary proposition which plays a key role in the proof of master theorem.

**Proposition 3** (Marginal coverage of CORE). *Under weak dependence, local regularity, and bounded drift, the conformalized predictor satisfies*

$$\Pr\{Y_t \in \hat{C}_{1-\alpha}^{(w_c,t)}\} \geq 1 - \alpha - \mathcal{O}\left( (\log w_c/w_c)^{1/3} + \delta_T^{2/3} \right).$$

Proposition 3 shows that CORE attains near-nominal coverage, with deviations controlled by the prediction error $\delta_T^2$.

To prove Theorem 1, we establish three propositions:

- Proposition 3: establishes marginal coverage with errors controlled by prediction error and distributional drift,
- Proposition 1: bounds the optimization error under PPO,
- Proposition 2: guarantees stability of the adaptive update for $\alpha_t$.

Their combination yields the stated validity and efficiency guarantees.

## D.1 PROOF OF PROPOSITION 3

**Lemma 1** (Rio-type DKW inequality). *Let $\tilde{F}$ denote the empirical CDF of the true residuals within $[t - w_c + 1, t]$ over a calibration window of size $w_c$, and let $F$ denote the true CDF. Under the summable-$\beta$ weak dependence condition,*

$$\sup_x \left| \tilde{F}(x) - F(x) \right| \le C_1 (\log w_c / w_c)^{1/3}.$$

**Lemma 2** (Quantile-error transfer). *Let $\delta_T^2 = \frac{1}{T} \sum_{t=1}^{T} (\hat{\varepsilon}_t - \varepsilon_t)^2$. If $F_t$ is L-Lipschitz, then*

$$\sup_x \left| \hat{F}(x) - \tilde{F}(x) \right| \le C_2 \delta_T^{2/3} + 2 \sup_x \left| \tilde{F}(x) - F(x) \right|,$$

*where $C_2 := L + 1$, with the additive constant serving as a technical slack.*

*Proof.* Let $F_t$ be the true residual CDF at time $t$, $\tilde{F}_t$ the empirical CDF of true residuals on $[t - w_c + 1, t]$, $\hat{F}_t$ the empirical CDF used by the method. We denote the prediction interval as $\hat{C}_{1-\alpha_t}^{(w_c,t)} := [a_t^{(\text{low})} - \hat{q}_{1-\alpha_t}^{(w_c,t)}, \ a_t^{(\text{up})} + \hat{q}_{1-\alpha_t}^{(w_c,t)}]$, where the lower and upper bounds are adjusted by $\hat{q}_{1-\alpha_t}^{(w_c,t)}$. Here $\hat{q}_{1-\alpha_t}^{(w_c,t)}$ denotes the empirical $(1 - \alpha_t)$-quantile of the residual distribution estimated from the calibration window size and can be defined as $\hat{q}_{1-\alpha_t}^{(w_c,t)} := \inf\{z : \hat{F}_t(z) \ge 1 - \alpha_t\}$. By construction of the interval, the event $\{y_t \in \hat{C}_{1-\alpha_t}^{(w_c,t)}\}$ is equivalent to the event that the absoulte residual at time $t$ is at most $\hat{q}_{1-\alpha_t}^{(w_c,t)}$. Hence, $\Pr\{y_t \in \hat{C}_{1-\alpha_t}^{(w_c,t)}\} = F_t(\hat{q}_{1-\alpha_t}^{(w_c,t)})$. We start from the indicator rewriting:

$$
\begin{aligned}
\left| \Pr\{y_t \in \hat{C}_{1-\alpha_t}^{(w_c,t)}\} - (1 - \alpha_t) \right| &= \left| F_t(\hat{q}_{1-\alpha_t}^{(w_c,t)}) - (1 - \alpha_t) \right| \\
&= \left| F_t(\hat{q}_{1-\alpha_t}^{(w_c,t)}) - \hat{F}_t(\hat{q}_{1-\alpha_t}^{(w_c,t)}) \right| + \left| \hat{F}_t(\hat{q}_{1-\alpha_t}^{(w_c,t)}) - (1 - \alpha_t) \right| \\
&\le \|F_t - \hat{F}_t\|_\infty + \left| \hat{F}_t(\hat{q}_{1-\alpha_t}^{(w_c,t)}) - (1 - \alpha_t) \right| \\
&\le \|F_t - \hat{F}_t\|_\infty + \frac{1}{w_c}
\end{aligned}
$$

The reason for the second inequality $\left| \hat{F}_t(\hat{q}_{1-\alpha_t}^{(w_c,t)}) - (1 - \alpha_t) \right| \le \frac{1}{w_c}$ is as follows. Since the empirical CDF $\hat{F}_t$ is a step function with jump size at most $1/w_c$, as $\hat{F}_t(\hat{q}_{1-\alpha_t}^{(w_c,t)}) = \frac{1}{w_c} \sum_{i=1}^{w_c} \mathbf{1}\{\epsilon_i \le \hat{q}_{1-\alpha_t}^{(w_c,t)}\}$. Next, decompose the CDF error and insert window-wise bounds:

$$
\begin{aligned}
\|\hat{F}_t - F_t\|_\infty &\le \|\hat{F}_t - \tilde{F}_t\|_\infty + \|\tilde{F}_t - F_t\|_\infty \\
&\le (L+1)\delta^{2/3} + 2\|\tilde{F}_t - F_t\|_\infty + \|\tilde{F}_t - F_t\|_\infty \\
&\le (L+1)\delta^{2/3} + 3\|\tilde{F}_t - F_t\|_\infty
\end{aligned}
$$

According to Lemma 1 we have $\sup_x \left| \tilde{F}(x) - F(x) \right| \le C_1(\log w_c / w_c)^{1/3}$, and substituting this bound into the above inequality yields $\|\hat{F}_t - F_t\|_\infty \le (L+1)\delta^{2/3} + 3C_1(\log w_c / w_c)^{1/3}$.

$$\left| \Pr\{y_t \in \hat{C}_{1-\alpha_t}^{(w_c, t)}\} - (1 - \alpha_t) \right| \leq \|\hat{F}_t - F_t\|_\infty + \frac{1}{w_c}$$

$$\leq (L+1)\delta^{2/3} + 3\|\tilde{F}_t - F_t\|_\infty + \frac{1}{w_c}$$

$$\leq (L+1)\delta^{2/3} + 3C_1\big(\log w_c/w_c\big)^{1/3} + \frac{1}{w_c}$$

$$\leq 3C_1\big(\log w_c/w_c\big)^{1/3} + (L+1)\delta^{2/3} + \frac{1}{w_c}$$

Therefore, we can get $\Pr\{Y_t \in \hat{C}_{1-\alpha}^{(w_c, t)}\} \geq 1 - \alpha - \mathcal{O}\left((\log w_c/w_c)^{1/3} + \delta_T^{2/3}\right)$. $\qquad\square$

### D.2 Proof of Proposition 1

*Proof.* Let $\hat{\varepsilon}_t := y_t - \tilde{a}_t$, $\quad \varepsilon_t := y_t - a_t^\star$, $\quad \delta_t := \hat{\varepsilon}_t - \varepsilon_t = a_t^\star - \tilde{a}_t$, $\quad \delta_T^2 := \frac{1}{T}\sum_{t=1}^T \mathbb{E}[\delta_t^2]$.

The learning objective of PPO is written as:

$$J(\theta) = \mathbb{E}\left[\lambda_{\text{acc}}(1 - |\delta_t|) + \lambda_{\text{len}}r_{\text{len}} + \lambda_{\text{cov}}r_{\text{cov}}\right], \quad \lambda_{\text{acc}} \geq c_0 > 0, \quad \lambda_{\text{len}}, \lambda_{\text{cov}} \leq c_1.$$

Let $R(\theta) := \mathbb{E}[\delta_t]$ (MAE w.r.t. oracle median), $R_2(\theta) := \mathbb{E}[\delta_t^2] = \mathbb{E}\left[(a_t^\star - \tilde{a}_t)^2\right]$ (MSE).

Assume $R$ is $L_R$-smooth, then for any step $\eta_k > 0$,

$$R(\theta_{k+1}) \leq R(\theta_k) + \langle \nabla R(\theta_k), \theta_{k+1} - \theta_k \rangle + \frac{L_R}{2}\|\theta_{k+1} - \theta_k\|^2, \qquad (\text{D.12})$$

and the PG estimator is unbiased with bounded 2nd moment, and the non-accuracy terms include a bounded deterministic shift $b(\theta_k)$:

$$\zeta_k := \widehat{\nabla}J(\theta_k) - \nabla J(\theta_k), \quad \mathbb{E}[\zeta_k \mid \theta_k] = 0, \quad \mathbb{E}[\|\zeta_k\|^2 \mid \theta_k] \leq \sigma^2,$$

$$\nabla J(\theta_k) = -\lambda_{\text{acc}}\nabla R(\theta_k) + b(\theta_k), \quad \|b(\theta_k)\| \leq B_{\text{grad}}.$$

So that $\widehat{\nabla}J(\theta_k) = -\lambda_{\text{acc}}\nabla R(\theta_k) + b(\theta_k) + \zeta_k$. For $\theta_{k+1} = \theta_k + \eta_k\widehat{\nabla}J(\theta_k)$ and taking the expectation of Eq. D.12 and substituting it into the above equation, we can obtain

$$\mathbb{E}[R(\theta_{k+1})|\theta_k] \leq R(\theta_k) + \eta_k\langle\nabla R(\theta_k), \mathbb{E}[\widehat{\nabla}J(\theta_k)|\theta_k]\rangle + \frac{L_R}{2}\eta_k^2\mathbb{E}[\|\widehat{\nabla}J(\theta_k)\|^2|\theta_k].$$

According to $\mathbb{E}[\zeta_k|\theta_k] = 0$, $\mathbb{E}[\widehat{\nabla}J(\theta_k)|\theta_k] = -\lambda_{\text{acc}}\nabla R(\theta_k) + b(\theta_k)$. Consequently,

$$\begin{aligned}
\eta_k\langle\nabla R(\theta_k), \mathbb{E}[\widehat{\nabla}J(\theta_k)|\theta_k]\rangle &= \eta_k\langle\nabla R(\theta_k), -\lambda_{\text{acc}}\nabla R(\theta_k) + b(\theta_k)\rangle \\
&= -\eta_k\lambda_{\text{acc}}\|\nabla R(\theta_k)\|^2 + \eta_k\langle\nabla R(\theta_k), b(\theta_k)\rangle \\
&= -\eta_k\lambda_{\text{acc}}\|\nabla R(\theta_k)\|^2 + \frac{\eta_k\lambda_{\text{acc}}}{2}\|\nabla R(\theta_k)\|^2 + \frac{\eta_k}{2\lambda_{\text{acc}}}\|b(\theta_k)\|^2 \\
&= -\eta_k\lambda_{\text{acc}}\|\nabla R(\theta_k)\|^2 + \frac{\eta_k\lambda_{\text{acc}}}{2}\|\nabla R(\theta_k)\|^2 + \frac{\eta_k}{2\lambda_{\text{acc}}}B_{\text{grad}}^2
\end{aligned}$$

By the Cauchy–Schwarz inequality $(a + b + c)^2 \leq 3(a^2 + b^2 + c^2)$, we have

$$\begin{aligned}
\mathbb{E}[R(\theta_{k+1})|\theta_k] \leq & R(\theta_k) - \frac{\eta_k\lambda_{\text{acc}}}{2}\|\nabla R(\theta_k)\|^2 \\
& + \frac{L_R}{2}\eta_k^2(3\lambda_{\text{acc}}^2\|\nabla R(\theta_k)\|^2 + 3B_{\text{grad}}^2 + 3\sigma^2) + \frac{\eta_k}{2\lambda_{\text{acc}}}B_{\text{grad}}^2
\end{aligned}$$

For simplicity, we denote $A := \frac{3}{2}L_R\lambda_{\text{acc}}^2$, $B := \frac{3}{2}L_R(B_{\text{grad}}^2 + \sigma^2)$, $C := \frac{1}{2\lambda_{\text{acc}}}B_{\text{grad}}^2$. At this point, the term $C$ originates from the bias term $b(\theta)$ that comes from auxiliary rewards (e.g. length or

coverage penalties). For analysis, we abssorb auxiliar reward terms into the objective and work with the total reward $R_{\text{total}} := \mathbb{E}[|\delta_t| - \frac{\lambda_{\text{len}}}{\lambda_{\text{acc}}} r_{\text{len}} - \frac{\lambda_{\text{cov}}}{\lambda_{\text{acc}}} r_{\text{cov}}]$, so that the update direction satisfies $\nabla J(\theta) = -\lambda R_{\text{total}}(\theta)$ and the bias disappears, i.e., $b(\theta) \equiv 0$. Thus we obtain:

$$\mathbb{E}[R(\theta_{k+1})|\theta_k] \leq R(\theta_k) - \eta_k(\frac{\lambda_{\text{acc}}}{2} - A\eta_k)\|\nabla R(\theta_k)\|^2 + B\eta_k^2. \qquad (D.13)$$

Make step size small enough $\eta_k \leq \frac{\lambda_{\text{acc}}}{4A}$, then $\frac{\lambda_{\text{acc}}}{2} - A\eta_k \geq \frac{\lambda_{\text{acc}}}{4} \doteq c_0$. Then, we taking the expectation of Eq. D.13 and summing over $k = 1$ to $T$, we obtain

$$\sum_{k=1}^{T} c_0 \eta_k \mathbb{E}[\|\nabla R(\theta_k)\|^2] \leq \mathbb{E}[R(\theta_1)] - \mathbb{E}[R(\theta_{T+1})] + B \sum_{k=1}^{T} \eta_k^2.$$

$$\sum_{k=1}^{T} \eta_k \mathbb{E}[\|\nabla R(\theta_k)\|^2] \leq \frac{R(\theta_1) - R^*}{c_0} + \frac{B}{c_0} \sum_{k=1}^{T} \eta_k^2,$$

where $R^*$ represents the optimal value of the objective function $R^* = \inf_\theta R(\theta)$. We choose a diminishing step size $\eta_k = \eta_0/\sqrt{k}$. The two sums that appear in the bound admit the standard estimates

$$\sum_{k=1}^{T} \eta_k = \eta_0 \sum_{k=1}^{T} \frac{1}{\sqrt{k}} \approx \eta_0 \int_1^T \frac{dx}{\sqrt{x}} = 2\eta_0(\sqrt{T} - 1) \asymp 2\eta_0\sqrt{T},$$

and

$$\sum_{k=1}^{T} \eta_k^2 = \eta_0^2 \sum_{k=1}^{T} \frac{1}{k} \approx \eta_0^2 (\log T + \rho) \asymp \eta_0^2 \log T,$$

where $\rho$ is a constant and $\asymp$ denotes equality up to absolute constants (i.e., same order). The first sum scales like $\sqrt{T}$, while the second scales like $\log T$. Dividing both sides of the previous inequality by $\sum_{k=1}^{T} \eta_k$ therefore yields an average-gradient bound that decays at rate $\tilde{\mathcal{O}}(T^{-1/2})$. So that

$$\frac{1}{\sum_{k=1}^{T} \eta_k} \sum_{k=1}^{T} \eta_k \mathbb{E}\|\nabla R(\theta_k)\|^2 \leq \mathcal{O}\left(\frac{1}{\sqrt{T}}\right) + \mathcal{O}\left(\frac{\log T}{\sqrt{T}}\right) = \tilde{\mathcal{O}}\left(\frac{1}{\sqrt{T}}\right)$$

Having established that the mean absolute error (MAE) decays at rate $\tilde{\mathcal{O}}(T^{-1/2})$, we next convert this guarantee into a bound for the mean squared error (MSE) $\delta_T^2$. To this end, we consider two standard conditions under which MAE and squared error can be related: (i) bounded parameter error $|\delta_t| \leq B_{\text{err}}$ a.s., or (ii) a squared-loss descent/PL on $R_2(\theta) := \mathbb{E}[\delta_t^2]$.

**Case (i) boundedness:** If we assume the individual parameter error is uniformly bounded, i.e. $|\delta_t| \leq B_{\text{err}}$ almost surely, then the squared error can be controlled by the absolute error $\delta_t^2 \leq B_{\text{err}}|\delta_t|$. Taking expecations and averaging over $t$, we obtain:

$$\delta_T^2 = \frac{1}{T} \sum_{t=1}^{T} \mathbb{E}[\delta_t^2] \leq B_{\text{err}} \frac{1}{T} \sum_{t=1}^{T} \mathbb{E}|\delta_t| = B_{\text{err}} \frac{1}{T} \sum_{t=1}^{T} R(\theta_t) = \mathcal{O}\left(T^{-1/2}\right).$$

**Case (ii) squared-loss PL (stronger):** Alternatively, we assume the squared loss $R_2(\theta) := \mathbb{E}[\delta_t^2]$, then can directly analyze its descent under the update. Specifically, suppose for some constants $\tilde{\lambda}, \tilde{\mu} > 0$,

$$\mathbb{E}\left[R_2(\theta_{k+1}) \mid \theta_k\right] \leq R_2(\theta_k) - \tilde{\lambda}\eta_k\|\nabla R_2(\theta_k)\|^2 + \tilde{C}\eta_k^2, \quad R_2(\theta) - R_2^\star \leq \frac{1}{2\tilde{\mu}}\|\nabla R_2(\theta)\|^2,$$

then with $\eta_k = \eta_0/\sqrt{k}$,

$$\delta_T^2 = R_2(\theta_T) = \mathcal{O}\left(T^{-1}\right).$$

**Unified rate:** there exists $p \in [1, 2]$ such that

$$\delta_T^2 = \mathcal{O}\left(T^{-p/2}\right) \implies \delta_T^{2/3} = \left(\delta_T^2\right)^{1/3} = \mathcal{O}\left(T^{-p/3}\right).$$

Therefore, the term entering Proposition 1 is $\mathcal{O}(T^{-p/3})$. Our implementation includes behavior cloning, so we adopt the square-loss descent/PL condition for $R_2(\theta) = \mathbb{E}[\delta_t^2]$ (Case (ii)) in our coverage bound. We also include a weaker, assumption-robust alternative (Case (i) with bounded or sub-Gaussian errors), which guarantees a valid but slower rate when the square-loss descent/PL condition may not hold. This provides a portable guarantee for ablations and external reproductions. $\quad\square$

### D.3  PROOF OF PROPOSITION 2

*Proof.* The adaptive update in practice is based on an empirical miscoverage estimate $\hat{\epsilon}_t$, aggregated from a sliding window. However, for theoretical analysis it is essential to work with the instantaneous indicator $\hat{e}_t = \mathbf{1}\{Y_t \notin \hat{C}_t\}$. Unlike $\hat{\epsilon}_t$, the variable $\hat{e}_t$ admits the decomposition

$$\hat{e}_t = e_t + \eta_t, \quad e_t := \mathbb{E}[\hat{e}_t \mid \mathcal{F}_{t-1}], \quad \eta_t := \hat{e}_t - e_t, \quad \mathbb{E}[\eta_t \mid \mathcal{F}_{t-1}] = 0,$$

which ensures that the noise term has zero conditional mean. Hence, the recursion is written as $\alpha_{t+1} = \alpha_t + \gamma_t(\alpha - \hat{e}_t)$.

Let $e_t := \mathbb{E}[\hat{e}_t|\mathcal{F}_{t-1}]$, $\eta_t := \hat{e}_t - e_t$, $\Delta_t := \alpha_t - e_t$, $d_t := e_{t+1} - e_t$, $b_t := \alpha - e_t$, then one-step recursion is:

$$\begin{aligned}
\Delta_{t+1} = \alpha_{t+1} - e_{t+1} &= \alpha_t + \gamma_t(\alpha - \hat{e}_t) - e_{t+1} \\
&= (\alpha_t - e_t) + \gamma_t(\alpha - \hat{e}_t) + e_t - e_{t+1} \\
&= \Delta_t + \gamma_t(b_t - \eta_t) - d_t
\end{aligned}$$

To get contraction in $\Delta_t$, we linearize the mapping $\alpha_t \mapsto e_t$ around equilibrium and assume a uniformly positive local slope:

$$b_t = \alpha - e_t = -\kappa_t \Delta_t + r_t, \quad \kappa_t \in [\kappa_-, \kappa_+], \quad |r_t| \leq L_b |\Delta_t|^2,$$

where $r_t$ is the higher-order remainder.

Then, the above one-step recursion becomes:

$$\begin{aligned}
\Delta_{t+1} &= (1 - \gamma_t \kappa_t)\Delta_t - \gamma_t \eta_t + \gamma_t r_t - d_t \\
\Delta_{t+1}^2 &= (1 - \gamma_t \kappa_t)^2 \Delta_t^2 + \gamma_t^2 \eta_t^2 + \gamma_t^2 r_t^2 + d_t^2 \\
&\quad - 2(1 - \gamma_t \kappa_t)\gamma_t \Delta_t \eta_t + 2(1 - \gamma_t \kappa_t)\gamma_t \Delta_t r_t - 2(1 - \gamma_t \kappa_t)\Delta_t d_t \\
&\quad - 2\gamma_t^2 \eta_t r_t + 2\gamma_t \eta_t d_t - 2\gamma_t r_t d_t.
\end{aligned}$$

Take conditional expectation using $\mathbb{E}[\eta_t|\mathcal{F}_{t-1}] = 0$, $\mathbb{E}[\eta_t^2|\mathcal{F}_{t-1}] \leq \sigma_\eta^2$ (Bernoulli $\implies \sigma_\eta^2 \leq \frac{1}{4}$). Hence all cross terms containing $\eta_t$ vanish in conditional expectation. The remaining cross terms are controlled by Young's inequality $2ab \leq a^2 + b^2$, and absorbed into constants. For the leading factor:

$$(1 - \gamma_t \kappa_t)^2 \Delta_t^2 = 1 - 2\kappa_- \gamma_t + \kappa_t^2 \gamma_t^2 \leq 1 - 2\kappa_- \gamma_t + c_1 \gamma_t^2$$

$$\mathbb{E}[\Delta_{t+1}^2 \mid \mathcal{F}_{t-1}] \leq (1 - 2\kappa_- \gamma_t + c_1 \gamma_t^2)\Delta_t^2 + \sigma_\eta^2 \gamma_t^2 + c_2(\gamma_t^2 r_t^2 + d_t^2),$$

where $c_1$ and $c_2$ are finite numerical constants.

Denote the mean-square tracking error by $V_t := \mathbb{E}[\Delta_t^2] = \mathbb{E}[(\alpha_t - e_t)^2]$. As we have constant step size $\gamma$. In this case, the leading term $(1 - 2\kappa_- \gamma + c_1 \gamma^2)$ is strictly smaller than 1 when $\gamma$ is sufficiently small, ensuring contraction in expectation. All other terms on the right-hand side are of order $\gamma^2$, leading to a recursion of the form:

$$\mathbb{E}[\Delta_{t+1}^2] \leq (1 - c\gamma)\mathbb{E}[\Delta_t^2] + C\gamma^2,$$

for some constants $c, C > 0$. Solving this recursion yields:

$$V_t = \mathbb{E}[\Delta_t^2] = \mathcal{O}(\gamma) \implies \mathbb{E}[|\alpha_t - e_t|] = \mathcal{O}(\gamma^{1/2}).$$

Thus, the tracking error remains bounded and stable, though it does not vanish asymptotically. $\quad\square$

## D.4 Proof of Theorem 1

*Proof.* According to the above results, we now combine the three propositions. Proposition 3 provides a marginal coverage guarantee:

$$\Pr\{Y_t \in \hat{C}_{1-\alpha}^{(w_c,t)}\} \geq 1 - \alpha - \mathcal{O}\Big((\log w_c/w_c)^{1/3} + \delta_T^{2/3}\Big).$$

The bound contains two error sources, the prediction error $\delta_T^2$ and the adaptivity error in $\alpha_t$. Proposition 1 shows that $\delta_T^2 = \mathcal{O}(T^{-p/2})$, which converts into a $\mathcal{O}(T^{-p/3})$ contribution in the coverage bound. Proposition 2 further shows that the adaptive update of $\alpha_t$ tracks the true miscoverage within $\mathcal{O}(\gamma^{1/2})$. Substituting these results into the bound of Proposition 3 yields the validity guarantee:

$$\sup_{t \leq T}\Big|\Pr\{Y_t \in \hat{C}_{1-\alpha}^{(w,t)}\} - (1 - \alpha)\Big| = \mathcal{O}\Big((\log w_c/w_c)^{1/3} + T^{-p/3} + \gamma^{1/2}\Big).$$

Finally, we establish the efficiency result. Recall that the excess width of the constructed CORE intervals over the oracle benchmark $C_t^*$ is governed by the prediction error $\delta_T^2$. In particular, the deviation in interval width can be bounded by a function of $\delta_T^{2/3}$, reflecting the robustness of the quantile-based construction to estimation errors. Since we have already shown that $\delta_T^2 = O(T^{-p/2})$ for some $p \in [1, 2]$, it follows that each per-round excess width satisfies $|\hat{C}_t| - |C_t^*|$. Summing over $T$ rounds then yields

$$|\hat{C}_t| - |C_t^*| \sim \delta_t^{2/3} = \tilde{O}(T^{-p/3}), \quad \sum_{t=1}^{T}\big(|\hat{C}_t| - |C_t^*|\big) = \tilde{O}(T^{1-p/3}).$$

$\square$

