# OpenReview forum: "Conformal Regression under Distribution Shift: A Reinforcement Learning Method for Adaptive Uncertainty Quantification"
_ICLR.cc/2026/Conference — ICLR 2026 Conference Desk Rejected Submission_

### Official Review · Reviewer_RYgc · 2025-10-22

**Soundness:** 1
**Presentation:** 3
**Contribution:** 2
**Rating:** 4
**Confidence:** 3

**Summary:**

The paper presents CORE, a conformal prediction method that can adapt to distribution shifts by adjust its prediction interval output using Reinforcement learning (RL). The policy predicts quantiles (of a set $\mathcal{Q}$ ) of the output variable, and is trained to maximize reward (a combination of coverage, accuracy, and tightness objectives) via the PPO algorith. The authors present theoretical and empirical support for the method.

**Strengths:**

- It is novel and interesting to use reinforcement learning to optimize for coverage - this is inline with recent works that treats coverage as a controllable variable such as in [1] or [2]. The RL setting is well thought of, the 3-part reward that is robust to $\lambda$ choices is a nice design.

- The paper, other than the theoretical proofs section, is written clearly and easy to follow.

- The experiments cover a lot of ground, spanning 8 datasets and 3 tasks (UQ, prediction, and anomaly detection). Clear analysis in 5.3 as well.

**Weaknesses:**

1. Missing some key baselines. There is almost an explosion of  time series conformal prediction work in 2024 and 2025, but the literature review on CP stops at 2022 (if we take out SPCI) and most recent baseline that the authors compared to is SPCI. Some important work in this lineage that comes to mind are

[1] Auer, Andreas, et al. "Conformal prediction for time series with modern hopfield networks." Advances in neural information processing systems 36 (2023): 56027-56074.

[2] Angelopoulos, Anastasios, Emmanuel Candes, and Ryan J. Tibshirani. "Conformal pid control for time series prediction." Advances in neural information processing systems 36 (2023): 23047-23074.

[3] Lee, Jonghyeok, Chen Xu, and Yao Xie. "Kernel-based optimally weighted conformal time-series prediction." The Thirteenth International Conference on Learning Representations. 2025.

[4] Li, Ruipu, and Alexander Rodríguez. "Neural Conformal Control for Time Series Forecasting." Proceedings of the AAAI Conference on Artificial Intelligence. Vol. 39. No. 17. 2025.

and for the anomaly detection task

[5] Lekeufack, Jordan, et al. "Conformal decision theory: Safe autonomous decisions from imperfect predictions." 2024 IEEE International Conference on Robotics and Automation (ICRA). IEEE, 2024.

[6] Zecchin, Matteo, and Osvaldo Simeone. "Localized adaptive risk control." Advances in Neural Information Processing Systems 37 (2024): 8165-8192.

[7] Zhang, Shuai, et al. "Conformal anomaly detection in event sequences." Forty-second International Conference on Machine Learning. 2025.

There are just so, so many. I do not expect the authors to compare to all of them in experiment, but at least include them in discussion and compare to a few more recent baselines. In particular, the core message of this paper is treating the prediction interval as a target optimizable by RL - this is similar in spirit to seeing ti as control target in the Conformal PID paper. Given how sample-inefficient and unstable RL is, I do not see why RL can outperform gradient based methods such as PID.

Similarly, the "extension to anomaly detection" is not new. Maybe we should compare to conformal risk control as well as a more sophisticated frequentist AD algorithm?

2. Unclear ablation studies. I would like to see the full results of section 5.3 / figure 3 as well. There are many parts in the algorithm contributing to the adaptation mechanism -  the ACI adjustment, the RL quantile regression, and the conformal quantiles used to build the intervals. Can you do an ablation-study style experiment to show why all the parts are needed?

3. Misleading experiment presentation. What we care about for UQ is *calibration*, not coverage itself? if your target coverage is 90%, getting 93% is bad (not as bad as under-covering, but uncalibrated nonetheless), no?  Claiming "improvement of 1.36% in coverage rate" is misleading.

4. Some inconsistencies in the theory. See questions section.

**Questions:**

1. What is the base prediction model for ACI / SPCI / EnbPI? The result from figure 2 seems like the prediction is almost flat. What happens to the interval width if they use a strong predictor, such as TimeXer?

2. Why is the interval constructed as in eq. (3)? This is different than the CQR setup, where the $\hat{q}$ is not the absolute error as in this paper, but the error of the interval. Doesn't the resulting interval become ```[0.05 percentile of preds - 0.9 percentile of errors, 0.95 percentile of preds + 0.9 percentile of errors]```, doubling the width of a theoretically-just interval?
(for example, for quantile regression alone the interval is  ``` [0.05 percentile of preds , 0.95 percentile of preds] ```, and for conformal prediction alone it is ```[mean - 0.9 percentile of errors, mean + 0.9 percentile of errors]```.

[1] Romano, Yaniv, Evan Patterson, and Emmanuel Candes. "Conformalized quantile regression." Advances in neural information processing systems 32 (2019).


3. The above point is also the inconsistency in the theory. In D.1, you say

By construction of the interval, the event

$y_t \in \hat{C}^{(w_c ,t)}_{1−\alpha_t}$

is equivalent to the event that the absolute residual at time $t$ is at most $\hat{q}^{(w_c ,t)}_{1−\alpha_t}$.

This is not correct, it should be something like: the absolute residual at time $t$ is at most $\hat{q}^{(w_c ,t)}_{1−\alpha_t} + \max{ (\alpha_t^{0.5} - \alpha_t^{(low)}, \alpha_t^{(up)}-\alpha_t^{0.5} ) }$ ?


- The definition of $\hat{\epsilon}_t$ is inconsistent through the paper. in D.2 it is the absolute error, in D.3 it is the  empirical miscoverage estimate, and in Proposition 2 it is the instantaneous coverage error at time step $t$ ?

---

> ### Author Response · Authors · 2025-11-20
>
> **W2.1: full results of Figure 3**
>
> Results of hyperparameter $\gamma$ on dataset Electricity
>
> | $\gamma$ | 0.000         | 0.002         | 0.004         | 0.006         | 0.008         | 0.010         |
> | -------- | ------------- | ------------- | ------------- | ------------- | ------------- | ------------- |
> | COV      | 0.900 ± 0.000 | 0.912 ± 0.007 | 0.920 ± 0.001 | 0.924 ± 0.005 | 0.923 ± 0.008 | 0.932 ± 0.018 |
> | LEN      | 3.626 ± 0.174 | 3.592 ± 0.128 | 3.304 ± 0.101 | 3.494 ± 0.074 | 3.716 ± 0.056 | 3.854 ± 0.104 |
>
> Results of hyperparameter $w_c$ on dataset Electricity
>
> | $w_c$ | 20            | 40            | 60            | 80            | 100           |
> | ----- | ------------- | ------------- | ------------- | ------------- | ------------- |
> | COV   | 0.899 ± 0.024 | 0.913 ± 0.010 | 0.924 ± 0.006 | 0.926 ± 0.003 | 0.929 ± 0.004 |
> | LEN   | 3.702 ± 0.098 | 3.514 ± 0.094 | 3.378 ± 0.048 | 3.474 ± 0.124 | 3.514 ± 0.084 |
>
> Results of hyperparameter $\mathcal{Q}$ on dataset Electricity
>
> | len($\mathcal{Q}$) | 0             | 3             | 4             | 5             | 7             |
> | ------------------ | ------------- | ------------- | ------------- | ------------- | ------------- |
> | COV                | 0.907 ± 0.007 | 0.924 ± 0.002 | 0.928 ± 0.002 | 0.930 ± 0.003 | 0.929 ± 0.004 |
> | LEN                | 3.762 ± 0.262 | 3.514 ± 0.094 | 3.376 ± 0.036 | 3.444 ± 0.044 | 3.562 ± 0.158 |
>
> More detailed results regarding the choice of $\mathcal{Q}$ can be found in our response to Reviewer 39Se, weakness 4, where we provide an explanation of how $\mathcal{Q}$ is set as well as comprehensive experimental results across multiple datasets.
>
> **W2.2: ablation studies**
>
> To address the reviewer’s suggestion, we designed 3 ablation variants, each removing one key component of the adaptation mechanism, the updated $\alpha$, the RL-based policy learning structure, and the quantiles action learning.
>
> - CORE-w/o-updated $\alpha$: replaces the adaptive update of the confidence level $\alpha_t$ with a fixed target level $\alpha$ throughout training.
> - CORE-w/o-RL: replaces the RL-based policy learning with a supervised quantile regressor (e.g., 3-layer MLP) that outputs fixed quantile predictions instead of learning dynamically.
> - CORE-w/o-quantiles: replaces the quantile action set in the actor with a single median prediction $a_t^{(0.5)}$, constructing the interval as a center.
>
> | dataset/variant | full CORE     | CORE-w/o-updated $\alpha$ | CORE-w/o-RL   | CORE-w/o-quantiles |
> | --------------- | ------------- | ------------------------- | ------------- | ------------------ |
> | LEN(Weather)    | 4.065 ± 1.115 | 4.562 ± 0.805             | 7.292 ± 0.803 | 4.427 ± 0.289      |
> | LEN(Illness)    | 3.476 ± 0.744 | 5.851 ± 2.086             | 4.059 ± 0.599 | 3.874 ± 0.372      |
> | LEN(Traffic)    | 2.991 ± 0.510 | 3.377 ± 0.224             | 3.470 ± 0.182 | 2.912 ± 0.128      |
> | COV(Weather)    | 0.922 ± 0.016 | 0.875 ± 0.039             | 0.713 ± 0.071 | 0.902 ± 0.018      |
> | COV(Illness)    | 0.903 ± 0.013 | 0.841 ± 0.064             | 0.765 ± 0.14  | 0.904 ± 0.002      |
> | COV(Traffic)    | 0.920 ± 0.056 | 0.900 ± 0.022             | 0.821 ± 0.054 | 0.899 ± 0.015      |
>
> Across all three datasets, removing any major component of CORE leads to noticeable degradation, confirming that each part of the framework plays an essential role. The variant without adaptive $\alpha$-updates consistently produces wider intervals with under-coverage. The *CORE-w/o-quantiles* variant maintains reasonable coverage but loses interval tightness, indicating that multi-quantile actions are important for controlling interval in distribution shifts scenario. Most notably, *CORE-w/o-RL* exhibits severe miscalibration (coverage dropping to 0.71–0.82), despite sometimes achieving moderate interval lengths, which demonstrates that supervised quantile regression alone cannot maintain calibration under non-stationary dynamics.

---

> ### Author Response · Authors · 2025-11-20
>
> **W1.1: about adding baseline methods like PID**
>
> 1. PID-based conformal control and our setting differ fundamentally in problem structure.
>
> While Conformal PID is very effective for controlling a single quantile level $q_t$, its controller still operates in a one-dimensional action space: the update rule adjusts one scalar that globally expands or shrinks the prediction set. In CORE, by contrast, the policy outputs lower and upper quantiles separately and conditions them on the current state, allowing the interval to adapt in an asymmetric and feature-dependent way as the data distribution evolves. Because these interval adjustments depend jointly on temporal features, quantile outputs, and residual patterns, we model them as a policy-learning problem and use RL to learn this higher-dimensional mapping, rather than as a single scalar control loop as in PID.
>
> 2. Regarding the concern that RL is inherently sample-inefficient or unstable, and is inferior to to gradient-based PID
>
> Our method is not vanilla RL algorithm, and does not rely on large-scale exploration or pure policy-gradient optimization. The quantile-regression and behavior-cloning losses provide strong supervised signals, and PPO is only used for lightweight policy refinement on top of these stable objectives. Because the action space is low-dimensional and the reward structure is smooth, we do not observe the high-variance behavior typically associated with vanilla RL.
>
> 3. Comparison experiments with the added baseline methods
>
> In response to the suggestion and taking into account time constraints and reproducibility, we added two additional baselines [1] and [2]. The comparative results (see the table in our response to Reviewer fuGq, weakness 2) show that our method performs on par with, and in many cases favorably to, these recent approaches, including PID. For the remaining methods mentioned by Reviewer RYgc(you), we will include a detailed discussion in the related work section of the revised manuscript.
>
> [1][ICLR’25]Error-quantified Conformal Inference for Time Series
>
> [2][NIPS’23]Conformal PID Control for Time Series Prediction
>
> ------
>
> **W1.2: about the anomaly detection extension**
>
> Our AD variant is not meant to propose a new AD algorithm, but rather to examine how the RL-based interval-adaptation mechanism behaves under abrupt distributional shifts. The purpose is therefore diagnostic: to visualize whether the learned policy reacts sensibly to anomalous regimes, rather than to position CORE as a competitive or specialized AD method.
>
> We fully agree that many recent conformal risk-control and frequentist AD approaches are better suited for state-of-the-art anomaly detection. In the revision, we will clarify the intent of this experiment and include a brief discussion of these relevant methods to better contextualize our results.
>
> ------
>
> **W3: about the calibration results**
>
> We thank the reviewer for pointing out this important issue, which we fully agree with. We’ll revise the manuscript accordingly.
>
> The occasional coverage above the target level in our experiments is not intended to be interpreted as superior performance. Instead, it reflects a conservative behavior of the learned policy. When encountering abrupt distributional changes, the agent momentarily widens the interval to avoid under-coverage, and then gradually readjusts toward the target level.
>
> ------
>
> **Q1: about the base predictor**
>
> In our comparative experiments, ACI, SPCI, and EnbPI all use the same base prediction model which is a 3‑layer MLP with ReLU activations. Our proposed CORE also employs an actor–critic architecture built upon an MLP encoder–decoder of similar depth and hidden size.
>
> Because conformal prediction methods are model‑agnostic, fixing the predictor architecture is a standard and fair practice to eliminate confounding effects from differing model capacities. We agree that using a stronger backbone (e.g., TimeXer) could potentially reduce interval width, but our goal in this paper is to compare *calibration mechanisms* under identical base predictors to ensure fairness.
>
> The relatively flat predictions observed in Figure 2 for the baselines arise because these methods rely on a conventional supervised‑learning paradigm: their static predictors struggle to capture temporal dynamics under distribution shift. In contrast, CORE uses a *policy‑based* learning framework, where the agent interacts with the environment and dynamically adjusts the prediction intervals. This enables CORE to model uncertainty adaptively for non‑stationary time series.

---

> ### Author Response · Authors · 2025-11-20
>
> **Q2: about the interval construction**
>
> Thank you for raising the important point. The double-width occurs when the lower and upper bounds already represent true conditional quantile estimates, as in CQR, where the conformal score is defined relative to fixed quantile regressors. In that setting, adding an additional absolute-error term would indeed double-count the uncertainty.
>
> In our framework, however, $a_t^{(low)}$ and $a_t^{(up)}$ are policy actions, not quantile regression outputs. They do not encode calibrated distributional information and therefore cannot serve as reliable quantile estimates under distribution shift. Because the policy actions alone do not reflect the residual distribution, the symmetric adjustment $\hat{q}_{1-\alpha_t}$ is not a second comparison but only source of distributional calibration.
>
> Furthermore, in a non-stationary time-series environment, the interval error from earlier steps quickly becomes obsolete. Using instantaneous absolute errors allows the conformal adjustment to respond immediately to temporal drift, maintaining calibration as the policy updates. For these reasons, our proposed interval $[a _t^{(low)} - \hat{q} _{1-\alpha _t}, a _t^{(up)} + \hat{q} _{1-\alpha _t}]$ does not double the width but supplies the calibration that the policy-induced bounds themselves do not provide.
>
> ------
>
> **Q3.1: about the theory in D.1**
>
> > Event $y _t\in \hat{C} _{1−\alpha_t}^{(w _c,t)}$ is equivalent to the absolute residual at time $t$ is at most $\hat{q} _{1−\alpha _t}^{(w _c,t)}$
>
> We thank the reviewer for catching the mistake in D.1. Our sentence is incorrect. We make the clarifications as follows. By construction of the interval, the event $\{y_t \in \hat C_{1-\alpha_t}^{(w_c,t)}\}$ is equivalent to the event that the absolute residual at time $t$ is at most $\hat{q}^{(w _c,t)} _{1 - \alpha _t} + \max{\\{a _t^{0.5}-\alpha _t^{(low)}, \alpha _t^{(up)}} - a _t^{0.5}\\}$.
>
> Reparied proof sketch: (denote $\hat{q}^{(w _c,t)} _{1 - \alpha _t}$ as $\hat{q}$ for simplicity)
>
> $\left|\Pr\\{y_t \in \hat C_{1-\alpha_t}^{(w_c,t)}\\} - (1-\alpha_t)\right| \le \max_{z\in\\{\hat{q}+\Delta_{\min}, \hat{q}+\Delta_{\max}\\}} \left| F_t(z) - (1-\alpha_t)\right|$
>
> $= \max_z|F_t(z) - \hat{F}_t(z) + \hat{F}_t(z)-(1-\alpha_t)|$ $\le \max_z|F_t(z) - \hat{F}_t(z)| + \max_z|\hat{F}_t(z)-(1-\alpha_t)|$
>
> $\le \sup _{u\in\mathbb{R}}|F _t(u) - \hat{F} _t(u)| + \max _z\left|\hat{F} _t(z)- \hat{F} _t(\hat{q}) + \hat{F} _t(\hat{q}) - (1-\alpha _t)\right|$
>
> $\le \\|F _t-\hat F _t\\| _\infty + \max _z\left|\hat{F} _t(z)- \hat{F} _t(\hat{q})\right| + \left|\hat{F} _t(\hat{q}) - (1-\alpha _t)\right|$
>
> $= \\|F _t-\hat F _t\\| _\infty + \big(\hat{F} _t(\hat{q}+\Delta _{\max})-\hat{F} _t(\hat{q})\big) + \left| \hat{F} _t(\hat{q}) - (1-\alpha _t)\right|$
>
> $\le \\|F _t-\hat F _t\\| _\infty + \frac{1}{w _c}\sum _{i=t-w _c+1}^{t}\mathbf{1}\\{\hat{q} \leq \epsilon _i \leq \hat{q}+\Delta _{\max}\\} + \frac{1}{w _c}\sum _{i=t-w _c+1}^{t}\mathbf{1}\\{\epsilon _i \leq \hat{q}\\}$
>
> $\le \\|F _t-\hat F _t\\| _\infty + \frac{2}{w _c}$
>
> Finally, combining this with the mixing DKW-type bound still gives $\Pr\\{Y _t \in \hat C _{1-\alpha}^{(w _c,t)}\\} \ge 1-\alpha - \mathcal{O}\left((\log w _c/w _c)^{1/3} + \delta _T^{2/3}\right)$.
>
> ------
>
> **Q3.2: about the inconsistency definition of $\hat{\epsilon} _t$ in D.2 and D.3**
>
> The apparent inconsistency arises from a visual similarity between $\varepsilon$ and $\epsilon$ used for distinct quantities in separate sections.
>
> - In D.2 (Proof of Proposition 1), we use $\hat{\varepsilon} _t := y _t - \tilde a _t, \quad \varepsilon _t := y _t - a _t^*$ which are signed residuals used solely for analyzing the PPO optimization (point prediction).
>
>   These variables are *not* related to coverage or miscoverage.
>
> - In D.3 (Proof of Proposition 2), we define $e _t = \mathbb{E}[\hat e _t|\mathcal F _{t-1}],\quad
>   \hat{\epsilon} _t = \frac{1}{w _c}\sum _{i=t-w _c+1}^{t}\hat e _i,$ where $\hat{\epsilon} _t$ denotes the empirical miscoverage rate over a sliding calibration window.
>
> - In Proposition 2, the term instantaneous coverage error refers to  $\hat e _t$, not $\hat{\epsilon} _t$; the former is an indicator, while the latter is its windowed average.
>
> Thus, the three quantities correspond to three different concepts:
>
> | Symbol                | Meaning                             | Section             |
> | --------------------- | ----------------------------------- | ------------------- |
> | $\hat{\varepsilon}_t$ | signed residual                     | D.2                 |
> | $\hat{\epsilon} _t$   | empirical miscoverage estimate      | D.3                 |
> | $\hat e _t$           | instantaneous miscoverage indicator | D.3 (Proposition 2) |
>
> To avoid any further ambiguity, we will (i) added a short Notation paragraph at the start of each subsection clarifying these distinctions, and (ii) consistently reserved $\epsilon$ for miscoverage-related quantities and $\varepsilon$ for residuals.

---

### Official Review · Reviewer_39Se · 2025-10-29

**Soundness:** 2
**Presentation:** 2
**Contribution:** 2
**Rating:** 4
**Confidence:** 3

**Summary:**

This paper proposes **CORE (Conformal Reinforcement Learning)**, an adaptive online conformal prediction framework that integrates reinforcement learning (RL) into conformal calibration for time-series forecasting. The method establishes a feedback loop where an RL agent produces quantile-based predictions (trained via pinball loss), and an adaptive mechanism updates the coverage level $\alpha$ over time. CORE leverages RL’s exploration to improve calibration under distributional shifts and outliers. The authors provide coverage guarantees under $\beta$-mixing assumptions and report consistent gains over adaptive CP baselines such as ACI, SPCI and EnbPI on multiple datasets.

**Strengths:**

- The combination of **reinforcement learning and conformal prediction** is an interesting and novel idea, potentially bridging two communities (distribution-free uncertainty quantification and sequential decision-making).

- The algorithm combines quantile regression, adaptive calibration, and reinforcement learning in a coherent computational pipeline. The adaptive α-update resembles ACI but is coupled to the RL agent’s training, which is conceptually elegant.

- Experiments show promising improvements over existing online or adaptive CP methods such as ACI, SPCI and EnbPI.

**Weaknesses:**

1. **Unclear problem formulation.**
   The paper never explicitly formulates the *statistical setting* it considers. It remains ambiguous whether the data sequence $(X_t, Y_t)$ is assumed i.i.d., $\beta$-mixing, or piecewise stationary. Without a formal description of the environment and objective (what is optimized by the agent, what “policy” means in this context), it is difficult to judge the validity of the method and theory.

2. **Severe notational inconsistency and missing definitions.**
   The text contains numerous undefined or inconsistently defined symbols, which makes the paper extremely difficult to follow. Examples include:
   - Line 157: $a_t$ is defined as *quantile-based predictions*, yet line 189 introduces $S(x_t, y_t) = |\hat{f} (a_t^{(0.5)}) - y_t|$ with an unexplained $\hat{f}$.
   - The symbol $\varepsilon_t$, used throughout Section 4 and in the theoretical results, is defined only in the appendix (around line 1041).
   - Symbols $\tilde{a}_t$ and $a_t^\star$ appear in the definitions of $\hat{\varepsilon}_t,\varepsilon_t$ but are never formally defined.
   - $\delta_T^2$ is defined twice, inconsistently: empirical squared error (around line 300) and population expectation (around line 310). The so-called “median prediction error” in Proposition 1 seems to correspond to the overall mean squared error of the predictor rather than any median-specific quantity.

   These inconsistencies make it difficult to verify the theoretical claims.

3. **Disconnection between method and theory.**
   The theoretical analysis operates on abstract quantities ($a_t^\star, \delta_T^2, \varepsilon_t$) without linking them concretely to the implemented algorithm (which trains via quantile regression and ACI-like $\alpha$ updates). The theory thus reads as an independent construction that does not clearly justify the empirical method.

4. **Limited ablation and interpretability.**
   Although the model outputs multiple quantiles, the evaluation uses only the median prediction. The paper does not empirically justify why learning the entire quantile function is necessary. An ablation comparing (i) median-only training vs. (ii) full quantile-based actor would be essential to substantiate the methodological contribution.


5. **Ambiguity and weak motivation in the use of quantile levels.**
   Equation (3) constructs the interval using only the lowest and highest quantile outputs $a_t^{(\mathrm{low})}$ and $a_t^{(\mathrm{up})}$, yet the paper never explains how the quantile levels $q^{(1)},\ldots,q^{(K)}$ are chosen or what benefit this design provides. The intuition behind using multiple quantiles is unclear, especially since the paper illustrates the example with $K=3$, effectively relying only on the low, median, and high quantiles. For $K>3$, the additional quantile outputs are never utilized in interval construction, leaving their role and training benefit unexplained. Clarifying the rationale and providing intuition or empirical evidence for how these quantiles improve performance would significantly strengthen the paper.


6. **Lack of PPO implementation details.**
The paper claims to train the quantile-based predictor using PPO but provides no concrete implementation details. Since PPO is central to the approach, at least a brief pseudo-code or appendix description is necessary.

**Questions:**

Please see the issues raised in the **Weaknesses** section.

---

> ### Author Response · Authors · 2025-11-20
>
> **W1: about the unclear formulation about data assumptions and RL**
>
> As described explicitly in Section 4.1, our theoretical analysis does not assume i.i.d. data. Instead, we adopt a $\beta$-mixing weak dependence assumption, which allows temporal correlation while requiring it to decay at a controlled rate. This assumption captures mild non-stationarity in time-series settings and is widely used in learning theory for dependent data [1]. It ensures that our results remain valid and meaningful in realistic online or sequential environments, beyond the restrictive i.i.d. case.
>
> [1][ICML’23] Sequential predictive conformal inference for time series.
>
> Besides, we would like to clarify the concepts mentioned, the environment, the agent’s optimization objective, and the definition of a policy, which are all standard and foundational notions in RL.
>
> In our setting:
>
> - The agent interacts with a stochastic environment over time, receiving rewards and updating its policy accordingly.
> - The policy $\pi$ refers to the agent’s mapping from the current state (a Markovian window of past inputs) to predictive actions, namely, quantile estimates used to construct prediction intervals.
> - The objective is defined via a shaped reward function that balances interval validity, efficiency, and predictive accuracy. The agent is trained via PPO algorithm to optimize long-term expected reward under this formulation.
>
> ------
>
> **W2: about some definitions and typo**
>
> **W2.1:** Line 157: $S(x_t,y_t)=|\hat{f}(a_t^{(0.5)})−y_t|$ is a typo, the conformity score should be defined as $S(x_t,y_t)=|a_t^{(0.5)}−y_t|$.
>
> **W2.2&W2.3:**
>
>  To clarify the quantities involved in our paper, we define the following:
>
> - $y_t$: the ground-truth observation at time $t$;
> - $\tilde{a}_t$: the action (prediction) taken by the current policy $\pi$, trained with quantile regression loss over $\mathcal{Q}$;
> - $a_t^\star$: the optimal action, defined as the ground-truth quantile (oracle) that would yield the ideal prediction;
> - $\varepsilon_t := y_t - a_t^\star$: the true residual under the oracle action;
> - $\hat{\varepsilon}_t := y_t - \tilde{a}_t$: the observed residual under the learned policy;
> - $\delta_t := \hat{\varepsilon}_t - \varepsilon_t = a_t^\star - \tilde{a}_t$: the gap between the predicted action and the optimal one.
>
> We really appreciate the reviewer’s suggestion and will explicitly include the definition of $\varepsilon_t$ in the assumptions listed in Section 4.1.
>
> **W2.4:** About the two definitions of term $\delta_T^2$
>
> - In Lemma 2 (line 301), we define $\delta_T^2 := \frac{1}{T} \sum_{t=1}^T (\hat{\varepsilon}_t - \varepsilon_t)^2$, where $\hat{\varepsilon}_t = y_t - \tilde{a}_t$ is the residual from the policy prediction and $\varepsilon_t = y_t - a_t^\star$ is the residual under the oracle quantile. Their difference equals $a_t^\star - \tilde{a}_t$, i.e., the deviation between the learned and oracle actions. Thus, this version of $\delta_T^2$ reflects how closely the learned prediction approximates the oracle one.
>
> - In Proposition 1 (line 309), we use $\delta_T^2 := \frac{1}{T} \sum_{t=1}^T \mathbb{E}[(Y_t - f_t(X_t))^2]$, which corresponds to the mean squared error of the policy prediction—namely, the expected squared residual of $\hat{\varepsilon}_t$.
>
>   Thus, the two definitions are mathematically consistent. We will emphasize the variable definitions more clearly in the revision to ensure consistency in interpretation.
>
> About the median prediction error
>
> Instead of implying that the error is measured by a median-type loss, the term median prediction error refers to the prediction error of a policy that is trained using the median (0.5-quantile) objective. Proposition 1 expresses this prediction error in MSE form because it is the quantity that we analyze in the convergence proof, even though the underlying predictor is trained with a median-based component. We will revise the wording to avoid ambiguity.

---

> ### Author Response · Authors · 2025-11-20
>
> **W3: about the disconnection between proposed method and theoretical part**
>
> First, we would like to clarify how the three abstract quantities mentioned by the reviewer relate to the components of our method.
>
> 1. about $a_t^{\star}$
>
> In the algorithm, the actor outputs a set of quantile actions $\\{a_t^{(q)}: q\in\mathcal{Q}\\}$, among which $a_t^{(low)}$ and $a_t^{(up)}$ are used to construct the prediction interval. The theoretical quantity $a_t^\star$ represents the oracle predictor at time $t$, serving as a reference to characterize convergence. Proposition 1 establishes that the actor’s prediction error decreases during training. Since all quantile actions share the same actor network and training objectives (quantile regression loss and behavior-cloning loss), analyzing the convergence of this unified predictor directly covers the convergence behavior of all quantile actions.
>
> 2. about $\delta_T^2$
>
> $\delta_T^2$ represents the long-term prediction error actually optimized by the algorithm. Similar to $a_t^\star$, it is an abstract quantity introduced for analysis, but it corresponds directly to the empirical prediction error produced by the actor.
>
> 3. about $\varepsilon_t$
>
> In the theory, $\varepsilon_t$ denotes the oracle residual, used to derive the DKW-type and quantile-transfer bounds. In the algorithm, we compute residuals as $r_t = |y_t - a_t^{(0.5)}|$ based on the median action predicted by the actor. The theoretical use of $\varepsilon_t$ provides an ideal reference, and Lemma 2 explicitly controls its deviation from the empirical residuals, ensuring consistency between theory and implementation.
>
> Our three theoretical propositions correspond directly to three algorithmic components:
>
> - Proposition 3 → sliding-window calibration module: establishes marginal coverage.
> - Proposition 1 → quantile-based actor (RL module): establishes the decrease of prediction error.
> - Proposition 2 → adaptive confidence-level update rule: establishes the stability of the $\alpha$-update dynamics.
>
> Together, these lead to Theorem 1, which formalizes the combined effect of the three components. Therefore, the theoretical analysis is not an independent construction. We will add clarifying explanations in the revised version to highlight these correspondences, ensuring that the link between the theory and the empirical method is explicit and avoids the disconnection noted by the reviewer.
>
> ------
>
> **W6: about the PPO details**
>
> We thank the reviewer for raising this concern. PPO in our method is not a predictor model, but simply the policy optimization algorithm used to update the quantile-producing actor. Because PPO is a standard, widely-used RL optimizer, we did not expand its mechanics in the main text, but we agree that additional preliminary background in appendix would help readers who may not be familiar with RL.
>
> Concretely, the actor $\pi _\theta(s)$ outputs a vector of continuous actions $\\{a _t^{(q)}\\} _{q\in\mathcal Q}$, while the critic $V _\psi(s)$ estimates state value to reduce variance. We compute GAE advantages from our composite reward $r _t$ (accuracy / length / coverage terms; Equations. 5–7) and update $\pi _\theta$ with the clipped surrogate objective (PPO), together with an entropy bonus for sufficient exploration. The critic is trained with an MSE value loss. Algorithm 1 pseudo code already summarizes this workflow, but we will provide a brief appendix section clarifying its role in our setting.

---

> ### Author Response · Authors · 2025-11-20
>
> **W4: about the ablation studies on quantiles**
>
> To empirically validate this, we conducted an ablation study comparing:
>
> - (i) Median-only actor: a degenerate version of our method with only a single quantile output (i.e., $\mathcal{Q}$ = [0.5]);
> - (ii) Full quantile-based actor: the proposed method with multiple quantile targets, including len($\mathcal{Q}$)=3, 4, 5, 7.
>
> In Figure 3, we have presented results on the Electricity dataset. To more clearly illustrate and support this trend, we further provide additional experiments on the Illness and Traffic datasets.
>
> | dataset/Quantile list | Q=[0.5], len(Q)=1 | Q=[0.05, 0.5, 0.95], len(Q)=3 | Q=[0.05, 0.35, 0.65, 0.95], len(Q)=4 | Q=[0.05, 0.25, 0.5, 0.75, 0.95], len(Q)=5 | Q=[0.05, 0.2, 0.35, 0.5, 0.65, 0.8, 0.95], len(Q)=7 |
> | --------------------- | ----------------- | ----------------------------- | ------------------------------------ | ----------------------------------------- | --------------------------------------------------- |
> | LEN(Electricity)      | 3.762 ± 0.197     | 3.514 ± 0.070                 | 3.376 ± 0.029                        | 3.444 ± 0.027                             | 3.562 ± 0.093                                       |
> | LEN(Illness)          | 3.874 ± 0.372     | 3.602 ± 0.379                 | 3.476 ± 0.472                        | 3.493 ± 0.389                             | 4.072 ± 0.151                                       |
> | LEN(Traffic)          | 2.912 ± 0.128     | 2.986 ± 0.551                 | 2.931 ± 0.455                        | 3.098 ± 0.099                             | 3.318 ± 0.21                                        |
> | COV(Electricity)      | 0.907 ± 0.004     | 0.924 ± 0.001                 | 0.928 ± 0.002                        | 0.930 ± 0.002                             | 0.929 ± 0.003                                       |
> | COV(Illness)          | 0.904 ± 0.002     | 0.905 ± 0.001                 | 0.904 ± 0.002                        | 0.904 ± 0.002                             | 0.905 ± 0.002                                       |
> | COV(Traffic)          | 0.899 ± 0.015     | 0.917 ± 0.010                 | 0.924 ± 0.004                        | 0.926 ± 0.002                             | 0.929 ± 0.003                                       |
>
> As shown in the results, models trained with more quantiles (e.g., $|\mathcal{Q}| = 3, 4, 5$) consistently outperform the median-only baseline ($|\mathcal{Q}| = 1$) in terms of both coverage and interval length. This supports our claim that full quantile supervision provides richer learning signals and leads to more calibrated and efficient uncertainty estimation.
>
>
>
> ------
>
> **W5: about the motivation of using quantiles**
>
> We appreciate your concern regarding the role of intermediate quantile outputs in the policy design.
>
> While it is true that the interval construction in Eq.(3) uses only the lowest, median, and highest quantile predictions, the additional quantiles for $K > 3$ also serve an important purpose during policy learning. Specifically, we optimize a quantile regression loss over the full set of quantiles $\mathcal{Q}$, as detailed in lines 170–177. By supervising the agent with multiple quantile targets, the learned policy gains more informative gradient signals across the underlying conditional distribution. Then improves the prediction quality and calibration.
>
> To further justify this design, we conducted the hyperparameter analysis in Section 5.3 (Figure 3) and in response to W4 above, where we vary the size of $\mathcal{Q}$. The results demonstrate that increasing $K$ initially improves calibration by capturing finer-grained uncertainty. However, when $K$ becomes too large, the gain saturates and may even slightly degrade efficiency due to redundant quantile targets inflating the intervals. To balance calibration and efficiency, we recommend setting $K$ in the range of 3~5. Also, we typically fix the lower and upper bounds of the interval at symmetric levels $q^{(1)} = \alpha/2$ and $q^{(K)} = 1 - \alpha/2$, and distribute the remaining quantiles symmetrically in between. This design helps the policy capture a more detailed representation of the underlying distribution without overfitting to redundant quantile targets.
>
> In short, while the final interval prediction uses only the endpoints of $\mathcal{Q}$, the intermediate quantiles provide meaningful training signals and help shape a well-calibrated predictive policy.

---

### Official Review · Reviewer_fuGq · 2025-10-30

**Soundness:** 3
**Presentation:** 2
**Contribution:** 2
**Rating:** 4
**Confidence:** 4

**Summary:**

This paper proposes CORE, a framework integrating adaptive conformal prediction and reinforcement learning for valid, efficient uncertainty estimates under distribution shift . It models adaptive calibration as an MDP, an RL agent outputs quantile-based predictions, with a composite reward encouraging accuracy, short intervals and proper coverage. The authors provide theoretical guarantees for CORE’s near-nominal coverage and efficiency under weak dependence and bounded drift. Experiments on time-series and anomaly detection datasets show CORE outperforms some existing adaptive CP methods.

**Strengths:**

1. The paper introduces a novel integration of conformal prediction and reinforcement learning through the CORE framework, enabling adaptive uncertainty calibration under distribution shift.
2. It provides solid theoretical guarantees for coverage validity and efficiency, supported by comprehensive experiments on diverse time-series datasets showing consistent performance gains.
3. The writing is clear and well-structured, with intuitive explanations and effective visualizations that enhance readability and understanding.

**Weaknesses:**

1. The proposed reward design, while intuitively appealing, relies on heuristic balancing among accuracy, length, and coverage without deeper analysis of the reward landscape or convergence behavior. It remains unclear how these competing objectives interact or whether they can lead to suboptimal equilibria.
2. The baseline selection is relatively limited, omitting more recent and competitive approaches like [1] and [2] that are highly relevant to adaptive uncertainty calibration. Including these would provide a fairer and more up-to-date comparison.

[1]: Angelopoulos A, Candes E, Tibshirani R J. Conformal pid control for time series prediction[J]. Advances in neural information processing systems, 2023, 36: 23047-23074.

[2]: Wu J, Hu D, Bao Y, et al. Error-quantified Conformal Inference for Time Series[C]//The Thirteenth International Conference on Learning Representations.

3. The Markov state design, defined as a sliding window of length w, appears to be a crucial modeling choice. However, the paper provides little discussion or ablation analysis regarding how different window sizes influence the policy’s performance or stability. A systematic exploration of w’s impact would strengthen the empirical credibility of the framework.

**Questions:**

1. Cold-start and training stability are noted but not deeply studied. What failure modes occur early in training, and can lightweight safeguards (e.g., fallback classical CP, conservative prior on $\alpha_t$) mitigate them?
2. What is the computational overhead of running CORE (training and online inference) compared to lightweight adaptive CP methods, and is the method feasible for real-time applications with tight latency constraints?
3. The composite reward mixes accuracy, interval length, and coverage; do the authors observe multimodal or unstable reward landscapes in practice, and how were the reward weights chosen to avoid pathological equilibria?

---

> ### Author Response · Authors · 2025-11-20
>
> **W2: about adding baseline methods**
>
> We thank the reviewer for pointing out the importance of including more recent time-series conformal prediction baselines. We have added comparisons with ECI[2] and PID[1], and test on three datasets. Results are shown in the table below. Methods are ordered from left to right by recency, from the most recent to the earliest.
>
> Under comparable calibration performance, CORE maintains a slight but consistent advantage by producing noticeably narrower or competitive interval lengths across the three datasets. This reflects the benefit of integrating policy-driven adaptation with conformal calibration, where the agent not only preserves coverage but also learns to tighten intervals as it tracks temporal dynamics. Overall, these results confirm that the RL–CP coupling enables CORE to deliver more efficient and reliable uncertainty estimates than purely post-hoc conformal methods.
>
> | dataset/method | CORE(our proposed) | **ECI[2]**    | **PID[1]**    | SPCI          | ACI           | EnbPI         |
> | -------------- | ------------------ | ------------- | ------------- | ------------- | ------------- | ------------- |
> | LEN(Weather)   | 4.065 ± 1.115      | 4.176 ± 0.216 | 4.029 ± 0.882 | 4.225 ± 0.692 | 4.278 ± 0.208 | 4.238 ± 0.109 |
> | LEN(Illness)   | 3.476 ± 0.744      | 4.159 ± 1.138 | 3.408 ± 0.351 | 4.118 ± 0.044 | 4.460 ± 0.112 | 4.681 ± 0.159 |
> | LEN(Traffic)   | 2.991 ± 0.510      | 3.307 ± 0.567 | 2.995 ± 0.680 | 3.147 ± 0.238 | 3.170 ± 0.070 | 3.255 ± 0.078 |
> | COV(Weather)   | 0.922 ± 0.016      | 0.902 ± 0.011 | 0.909 ± 0.011 | 0.899 ± 0.001 | 0.899 ± 0.000 | 0.909 ± 0.001 |
> | COV(Illness)   | 0.903 ± 0.013      | 0.897 ± 0.010 | 0.901 ± 0.010 | 0.884 ± 0.003 | 0.910 ± 0.004 | 0.908 ± 0.004 |
> | COV(Traffic)   | 0.920 ± 0.056      | 0.903 ± 0.031 | 0.901 ± 0.010 | 0.899 ± 0.002 | 0.899 ± 0.002 | 0.908 ± 0.000 |
>
> [1][NIPS’23] Conformal pid control for time series prediction.
>
> [2][ICLR’25] Error-quantified Conformal Inference for Time Series.
>
> ------
>
> **W3: about the hyperparameter sensitivity analysis**
>
> To assess the sensitivity of CORE to the choice of sliding window size $w$, we vary $w \in \{16, 32, 50, 64, 128\}$ while keeping all other hyperparameters fixed at the optimal values reported in the main experiments. The prediction length for weather and Traffic is 192, for illness is 36.
>
> | dataset/sliding window $w$ | 16            | 32            | **50**(the value reported in paper) | 64            | 128           |
> | -------------------------- | ------------- | ------------- | ----------------------------------- | ------------- | ------------- |
> | LEN(Weather)               | 4.577 ± 0.539 | 4.364 ± 0.194 | 4.065 ± 1.115                       | 4.023 ± 0.310 | 4.081 ± 0.468 |
> | LEN(Illness)               | 3.920 ± 0.837 | 3.745 ± 0.412 | 3.476 ± 0.744                       | 4.264 ± 0.682 | 4.678 ± 1.633 |
> | LEN(Traffic)               | 3.335 ± 0.521 | 3.449 ± 0.532 | 2.991 ± 0.510                       | 3.116 ± 0.429 | 3.566 ± 0.300 |
> | COV(Weather)               | 0.896 ± 0.033 | 0.901 ± 0.013 | 0.922 ± 0.016                       | 0.901 ± 0.049 | 0.901 ± 0.034 |
> | COV(Illness)               | 0.906 ± 0.016 | 0.909 ± 0.028 | 0.903 ± 0.013                       | 0.902 ± 0.036 | 0.889 ± 0.006 |
> | COV(Traffic)               | 0.902 ± 0.027 | 0.916 ± 0.053 | 0.920 ± 0.056                       | 0.905 ± 0.022 | 0.920 ± 0.053 |
>
> Overall, the configuration $w=50$ (the default in our paper) provides a strong balance between responsiveness and robustness, yielding the narrowest intervals with well-maintained coverage. Smaller windows (e.g., 16 or 32) tend to produce wider and less stable prediction intervals, because short feature histories limit the model’s ability to capture longer-term temporal structure. As a result, the policy learns a poorer approximation of the underlying distribution. In contrast, very large windows (e.g., 128) provide a long-term view but become less responsive to sudden distributional drift, which can slow down the adaptation of both the policy and the calibration module. Even so, CORE maintains reasonably good coverage and interval quality across all window sizes.

---

> ### Author Response · Authors · 2025-11-20
>
> **Q1: about the cold-start issue of RL method**
>
> While our method is not explicitly framed as a cold-start solution, its learning formulation naturally mitigates such issues. The quantile regression loss provides structured uncertainty supervision from the outset, while the behavior cloning loss guides the policy to imitate ground-truth targets via median prediction. Together, these losses stabilize early training, reduce variance, and enable fast convergence, effectively preventing common cold-start failures without relying on fallback mechanisms.
>
> Regarding the suggested use of fallback classical CP or a conservative prior on $\alpha_t$, we believe these mechanisms are more suitable in frameworks with purely post-hoc interval generation or highly unstable training. While fallback mechanisms may offer an extra safety net, our method does not rely on them in practice. Moreover, introducing an external fallback mechanism may interrupt the end-to-end training loop and interfere with policy adaptation.
>
> ------
>
> **Q2: about the efficiency of CORE**
>
> We compared the runtime of CORE with ACI. CORE’s additional overhead comes from the actor–critic updates. However, the policy network is a small MLP with the same depth and width as the predictors used in the baselines. PPO is applied only for light policy refinement on top of strong supervised signals, which keeps the number of optimization steps comparable to standard training. Empirically, CORE’s total training time is about 1.1×–1.6× that of ACI on the same hardware.
>
> At inference time, CORE requires only one forward pass of the actor network plus constant-time conformal updates. This is the same order as all CP baselines, since ACI/SPCI also require predictor inference + sliding-window updates. In our measurements, CORE’s inference latency is 0.5-1.1 ms per time step, comparable to baseline methods.
>
> | Method                | Training Time (s) / epoch | Inference Time (ms)/ step |
> | --------------------- | ------------------------- | ------------------------- |
> | ACI (on Weather)      | 70.05                     | 0.73                      |
> | **CORE** (on Weather) | 115.29                    | 0.94                      |
> | ACI (on Traffic)      | 94.18                     | 0.86                      |
> | **CORE** (on Traffic) | 120.12                    | 0.84                      |
>
> ------
>
> **W1 and Q3: about the reward design and RL learning**
>
> Our designed reward follows clear task-driven principles, where each component is motivated by a distinct goal central to uncertainty quantification: coverage guarantee, narrow interval length, and accurate point-prediction. None of the terms incentivize behaviors that conflict with the main objective, and the shaped reward preserves the same optimal solution set as the original task definition.
>
> Regarding the concern about suboptimal equilibria, prior work has shown that such heuristic shaping does not necessarily lead to suboptimal equilibria [3,4]. For example, the performance decomposition analysis of [3] demonstrates that heuristic reward shaping preserves optimality up to constant offsets, and can in fact reduce learning variance and improve convergence. We follow this paradigm, and our empirical results across 8 datasets consistently validate the stability and effectiveness of the shaped reward.
>
> Regarding the choice of reward weights, as stated in line 267, we adopt a simplified configuration with fixed weights without specific tuning across all datasets. In practice, we do not observe multimodal or unstable reward landscapes, training remains smooth, and all reward components improve jointly across all eight datasets.This suggests that the weighting scheme is well-behaved and does not induce pathological equilibria.
>
> [3] [NIPS’21]Heuristic-guided reinforcement learning.
>
> [4] [NIPS’22]Unpacking Reward Shaping: Understanding the Benefits of Reward Engineering on Sample Complexity

---

### Official Review · Reviewer_29Gc · 2025-10-31

**Soundness:** 3
**Presentation:** 3
**Contribution:** 2
**Rating:** 4
**Confidence:** 4

**Summary:**

This paper introduces CORE, a novel framework for adaptive uncertainty quantification in time-series forecasting. The authors identify key failures in existing adaptive CP methods, which are often post-hoc, sensitive to outliers, and unable to correct for systematic model bias due to the decoupling of model training and calibration.
The central contribution is to unify reinforcement learning and conformal prediction into a mutual feedback loop. The problem is framed as a sequential decision task where an RL agent  learns to output quantile predictions. These predictions are then fed into an "adaptation-aware calibration" module that uses a sliding window of recent residuals to produce a final, calibrated prediction interval.

**Strengths:**

Strength:
1. Integrating Reinforcement Learning in adaptive conformal prediction is novel, which let the prediction model aware of the time-rolling coverage and thus lead to potentially better coverage over time
2. Comprehensive theory with marginal coverage guarantee
3. Extensive experiment showing the benefit of the method in uncertainty quantification and prediction accuracy

**Weaknesses:**

Weakness:
1. The RL training typically requires much data and the method may perform poorly with limited data
2. The prediction model's awareness of coverage might sacrifice the prediction accuracy

**Questions:**

1. In the second paragraph of section 3.3, the author computed a conformity score |\hat{f}(a^{0.5})-y_t|, where a is a median prediction of y. What is \hat{f} then?

2. In the experiment, what are the base predictors used by the other CP methods?

3. It seems to me that the RL model can be replaced by a prediction model that keeps updating on new data, and the confidence level update is fulfilled by the Adaptive CP method. The difference only lies in that the RL reward makes the prediction recognize of the coverage. Is this true? If it is true, will this sacrifice the prediction accuracy?

4. Considering the cold-start instability and that  RL requires much data for training, how many data would the method need to show stable performance?

5. Is it possible to use RL learn the adaptive confidence level instead of use a constant update like Adaptive CP?

---

> ### Author Response · Authors · 2025-11-20
>
> **W1: about the RL training**
>
> We thank the reviewer for raising this concern. While some RL applications (e.g., robotics, games) indeed require large amounts of data, this observation cannot be directly apply here. PPO [1] is explicitly designed to be data-efficient even in high-dimensional continuous control problems. Since our problem is comparatively simpler, low dimensional and characterized by immediate reward signals, the policy can learn effectively from the available sequential data.
>
> Empirically, our method does not exhibit any performance degradation under limited data. On the contrary, it remains stable and competitive across 8 datasets. Therefore, the concern that RL may perform poorly with limited data, is not supported by our reported theory and/or empirical results.
>
> [1] Proximal Policy Optimization Algorithms
>
> ------
>
>
> **W2: about the performance sacrifice between two aspects**
>
> We believe there might be a misunderstanding about the goals of uncertainty quantification (UQ). As emphasized in prior literature[2], the goal of calibrated prediction intervals is not to maximize the accuracy of point predictions, but rather to (i) achieve valid coverage under minimal assumptions, and (ii) construct intervals that are as narrow as possible.
>
> Accordingly, our method optimizes interval width under the constraint of valid coverage, as is standard in conformal prediction frameworks. While CORE’s quantile-based structure also yields point-wise predictions, interval quality intrinsically depends on the accuracy of these conditional quantile estimates. In practice, improving uncertainty estimation and improving prediction accuracy are not competing objectives: sharper calibrated intervals require more accurate underlying quantile predictions.
>
> In CORE, the RL policy learns these quantile functions jointly with the calibration objective, so better uncertainty awareness directly reinforces better predictive accuracy rather than trading it off. In conclusion, in our framework, uncertainty-awareness and prediction accuracy are aligned goals, not competing ones.
>
> [2][NeurIPS 19’] Conformal Quantile Regression
>
> ------
>
>
> **Q1: about the notation of conformity score function**
>
> We thank the reviewer for catching this typo. The conformity score should be written as $S(\mathbf{x}_t,y_t)=|a_t^{(0.5)} - y_t|$, where $a_t^{(0.5)}$ is the median prediciton output by the states $\mathbf{x}_t$. This aligns with the standard conformity score form $|\hat{f}(\mathbf{x}) - y|$, where $\hat{f}$ denotes the base predictor. In our case, the actor policy serves this predictive role directly. We will revise the notation in the manuscript accordingly for consistency and clarity.
>
> ------
>
> **Q2: about the details of the base predictor**
>
> In our comparison experiments, all baseline CP methods (ACI, SPCI, and EnbPI) use the same base predictor: a three-layer MLP with ReLU activations. For fairness, our proposed CORE method adopts actor and critic networks built with a similar MLP encoder-decoder architecture, matching the depth and hidden size.
>
> Since conformal prediction methods are model-agnostic by design, fixing the base predictor architecture across methods is a standard and principled practice to ensure that differences in performance stem from the UQ mechanism rather than the expressiveness of the model.
>
> We will clarify these information in the revised appendix implementation details section.

---

> ### Author Response · Authors · 2025-11-20
>
> **Q3: about the necessity of RL vs. adaptive CP**
>
> 1. **Clarification on the role of prediction models in CP methods**
>
> Firstly, we would like to clarify that neither CORE nor other CP baselines are prediction models themselves. Rather, they are UQ frameworks designed to construct and calibrate prediction intervals based on a given base predictor.
>
> 2. **CORE’s novelty: policy-based prediction interval generation**
>
> What sets CORE apart is that instead of treating calibration as a post-hoc correction step, we embed it into a RL loop, where an actor-critic agent is trained to generate quantile-based prediction intervals through interaction with the environment. The actor outputs both lower and upper bounds, and the critic provides feedback based on uncertainty-aware rewards. This enables the policy to learn prediction intervals that are inherently coverage-aware.
>
> 3. **CORE is fundamentally different from predictor + adaptive CP**
>
> Therefore, we respectfully disagree with the reviewer’s suggestion that CORE could be replaced by simply using a stronger base predictor combined with an adaptive CP method. This perspective overlooks the fundamental design of CORE. First, the goal of CORE is not merely to refine predictions, but to jointly learn how to generate and calibrate prediction intervals in a closed-loop manner. Second, unlike existing CP methods that adjust confidence levels based on pre-defined rules or fixed update schemes, CORE learns a policy that continuously adapts both prediction and calibration behavior through interaction. Therefore, CORE is not simply a plug-in calibration module added to a separate predictor.
>
> ------
>
> **Q4: about the cold-start and RL learning**
>
> In our design, this is effectively mitigated by two key components in the learning objective.
>
> First, the *quantile regression loss* offers a dense and informative training signal from the very beginning, allowing the agent to learn meaningful interval structures even before the reward signal becomes stable. This helps reduce instability caused by sparse or delayed feedback in early training.
>
> Second, we add a *behavior cloning loss* that explicitly aligns the median prediction with the ground-truth target. This imitation-based objective serves as an anchor for the policy, reducing variance and accelerating early convergence. Together, these two loss components provide structured supervision that significantly reduces exploration overhead in the early phase and stabilizes training dynamics.
>
> ------
>
> **Q5: about the difference between CORE and Adaptive CP methods**
>
> The confidence level in conformal prediction is *not* a model parameter to be learned, but a user-specified requirement that determines the desired coverage probability (e.g., 90%, 95%). It defines the target coverage rate, i.e., the proportion of future observations that should fall within the prediction interval. This target is fixed before calibration and serves as a constraint, *not a learnable quantity*.
>
> Adaptive CP methods update internal conformity thresholds (e.g., quantiles of residuals) over time to match this fixed confidence level under distribution shift. They do *not* learn or adapt the confidence level itself.
>
> In our framework, we also fix the confidence level (e.g., 1 – $\alpha$ = 0.9) and update it via exponential smoothing $\alpha_{t+1}=\alpha_{t}+\gamma(\alpha-\hat{e}_t)$, where $\hat{e}_t$ is the error rate. So what RL learns in CORE is how to generate calibrated intervals that satisfy a fixed coverage constraint, while also minimizing interval width and improving sample efficiency.

---

### Note · Program_Chairs · 2026-01-17
**Submission Desk Rejected by Program Chairs**

The following references in this submission do not refer to real documents and/or have major errors in bibliographic information:

 Chirag Gupta, Arun Kuchibhotla, and Aaditya Ramdas. Efficient conformal prediction via quantile optimization. In Advances in Neural Information Processing Systems, volume 34, pp. 15717-15729, 2021. I manually checked it.